# Intrinsic mechanical sensitivity of mammalian auditory neurons as a contributor to sound-driven neural activity

**Maria C Perez-Flores[1], Eric Verschooten[2], Jeong Han Lee[1], Hyo Jeong Kim[1], Philip X Joris[2], Ebenezer N Yamoah[1]\***

[1]Department of Physiology and Cell Biology, University of Nevada, Reno School of Medicine, Reno, United States; [2]Laboratory of Auditory Neurophysiology, Medical School, Campus Gasthuisberg, University of Leuven, Leuven, Belgium

**Abstract** Mechanosensation – by which mechanical stimuli are converted into a neuronal signal – is the basis for the sensory systems of hearing, balance, and touch. Mechanosensation is unmatched in speed and its diverse range of sensitivities, reaching its highest temporal limits with the sense of hearing; however, hair cells (HCs) and the auditory nerve (AN) serve as obligatory bottlenecks for sounds to engage the brain. Like other sensory neurons, auditory neurons use the canonical pathway for neurotransmission and millisecond-duration action potentials (APs). How the auditory system utilizes the relatively slow transmission mechanisms to achieve ultrafast speed, and high audio-frequency hearing remains an enigma. Here, we address this paradox and report that the mouse, and chinchilla, AN are mechanically sensitive, and minute mechanical displacement profoundly affects its response properties. Sound-mimicking sinusoidal mechanical and electrical current stimuli affect phase-locked responses. In a phase-dependent manner, the two stimuli can also evoke suppressive responses. We propose that mechanical sensitivity interacts with synaptic responses to shape responses in the AN, including frequency tuning and temporal phase locking. Combining neurotransmission and mechanical sensation to control spike patterns gives the mammalian AN a secondary receptor role, an emerging theme in primary neuronal functions.

**\*For correspondence:**
enyamoah@gmail.com

**Competing interest:** The authors declare that no competing interests exist.

## Editor's evaluation

The transduction of mechanical sound signals into electrical activities has so far been attributed to the inner hair cells of the inner ear. Here the authors show that the spiral ganglion neurons that innervate the inner hair cells are mechanosensitive as well. In particular, the authors find that spiral ganglion cells maintained in cell culture can fire action potentials in response to mechanical stimulation. Moreover, the authors provide evidence that the mechanotransduction in the spiral ganglion cells may contribute to sound detection in vivo.

## Introduction

The senses dependent on mechanosensation – hearing, balance, and touch – excel in speed and wide-ranging sensitivity among the sensory systems. The tactile sense has evolved beyond detecting simple mechanical stimuli to encode complex mechanical texture and vibration via unmyelinated and myelinated nerve endings and specialized mechanoreceptors (*Delmas et al., 2011*). For example, Pacinian corpuscles in mammalian skin encode vibrations up to ~1000 Hz – more than an order of magnitude

above the visual system's flicker-fusion threshold, which sets the limit for stable viewing of computer screens and fluid motion in moving images. Transduction's temporal acuity is directly translated into a neural code in such tactile receptors because transduction occurs in the same neural element that conducts the signal to the brain. This is different in the auditory and vestibular systems, where mechanosensation and messaging to the brain is subserved by separate cells (hair cells [HCs] and primary neurons). Their synaptic interface is a potential limit on temporal acuity.

Nevertheless, the vestibular system produces one of the fastest human reflexes, with a delay of only ~5 ms (*Huterer and Cullen, 2002*). It is thought that non-quantal neurotransmission in the huge calyceal synapse between vestibular HCs and first-order neurons is essential to this speed (*Eatock, 2018*). Paradoxically, a specialized mechanism observed in the tactile and vestibular systems has not been identified in hearing. Nevertheless, it is the mechanosensitive system for which temporal acuity reaches its highest limits. For example, auditory brainstem neurons can reliably code tiny interaural time and intensity differences toward spatial hearing (*Yin et al., 2019*) over an enormous range of frequencies and intensities. The acknowledged mechanism for activation of classical action potentials (APs) in the primary auditory nerve (AN), also called spiral ganglion neurons (SGNs), is neurotransmission at the HC-ribbon synapse. Stereociliary bundles of auditory HCs convert sound-induced displacement and depolarization to neurotransmitter release unto AN afferents (*Fettiplace, 2017*; *Fuchs et al., 2003*; *Roberts et al., 1988*) with notable speed and temporal precision. The ribbon-type synapse is equipped to sustain developmentally regulated spontaneous activity (<100 Hz) (*Levic et al., 2007*), sound-evoked APs (<~300 Hz), and phase-locked responses at auditory frequencies up to ~5 kHz (*Johnson, 1980*). Phase locking to sound stimuli is a feature of the AN essential for sound detection, localization, and arguably for pitch perception and speech intelligibility (*Peterson and Heil, 2020*; *Yin et al., 2019*). How these response features remain sustained, despite the limits of presynaptic mechanisms of transmitter release to ATP-generation, synaptic fatigue, and vesicle replenishment (*MacLeod and Horiuchi, 2011*; *Rutherford et al., 2021*; *Stevens and Wesseling, 1999*; *Yamamoto and Kurokawa, 1970*) are not fully understood.

Responses of SGNs consist of multiple components, but the underlying mechanisms remain unresolved. Inner HC (IHC) depolarization increases the frequency of excitatory postsynaptic current (EPSC). Moreover, the synaptic current amplitude remains unchanged (*Glowatzki and Fuchs, 2002*; *Grant et al., 2010*; *Siegel, 1992*). In response to low-frequency tones, the HC-ribbon synapse triggers APs over a limited phase range of each sound cycle. Low- and high-intensity tones elicit phase-locked AN responses, which generate unimodal cycle histograms, that is, response as a function of stimulus phase (*Johnson, 1980*; *Rose et al., 1967*). In contrast, for some intermediate intensities, AN fibers fire APs at two or more stimulus phases referred to as 'peak-splitting' (*Johnson, 1980*; *Kiang and Moxon, 1972*). Moreover, a typical AN frequency tuning curve consists of two components (*Liberman and Kiang, 1984*) – a sharply tuned tip near the characteristic frequency (CF) and a low-frequency tail, which are differentially sensitive to cochlear trauma. These observations suggest that more than one process may drive AN responses. Whereas the contribution of fast synaptic vesicle replenishment at the HC-ribbon synapse is clear (*Griesinger et al., 2005*), the sustained nature of the IHC-AN synapse to sound (*Glowatzki and Fuchs, 2002*; *Griesinger et al., 2005*; *Li et al., 2014*) also raises the possibility that multiple processes sculpt AN responses. Utilization of the mechanical energy dissipated from sound-induced cochlear motion is an attractive model since neurons are by and large mechanically sensitive (*Gaub et al., 2020*).

We found that SGNs respond to minute mechanical displacement. Motivated by this finding, we hypothesized that SGN afferents actively sense the organ of Corti (OC) movement (OCM) and that this sensitivity, together with neurotransmission, shapes AN properties. The geometry of the course of the unmyelinated terminal segment of SGN dendrites toward the OC suggests that this segment undergoes some degree of mechanical deformation in response to sound. These dendrites emanate from the habenula perforata and angle toward their IHC target between the osseous spiral lamina, basilar membrane, and inner pillar cells (*Lim, 1986*). The OC is thought to rotate about a pivot point near the tip of the osseous spiral lamina. An attractive feature of the SGN mechanical-sensitivity hypothesis is that it provides a straightforward substrate for one of two interacting pathways thought to generate peak-splitting and/or multi-component frequency tuning (*Liberman and Kiang, 1984*). Using in vitro simultaneous whole-cell recordings and mechanical stimulation, we show that adult SGNs are mechanically sensitive. Mechanical stimulation of the cell body (soma) elicits an inward current, which

reverses at ~0 mV. Mechanically activated (MA) inward currents ($I_{MA}$) and membrane voltage responses are sensitive to GsMTx4 peptide, a mechanosensitive channel blocker (*Bae et al., 2011*). Mechanical stimulation of the soma and dendrites elicits adapting, bursting, or non-adapting firing responses. Simultaneous sinusoidal stimulation with current injection and mechanical displacement alters firing rate and temporal coding. In vivo, single-unit recordings from AN fibers demonstrate that 2,3-dihydroxy-6-nitro-7-sulfamoyl-benzo[f]quinoxaline (NBQX) and $Ca^{2+}$ channel blockers, potent inhibitors of synaptic transmission, suppress spontaneous APs, but sound-evoked APs persist at high intensities. These findings suggest that AN fibers' intrinsic mechanical sensitivity contributes to sound-evoked activity and ill-understood features such as multi-component AN tuning curves and peak-splitting. Our results also indicate that primary neurons may support sensory receptor functions, an emerging notion (*Hattar et al., 2002*; *Woo et al., 2014*).

## Results

### Mouse auditory neurons respond to stepped mechanical stimulation

*Figure 1a* shows the responses of adult mouse SGNs subjected to two different bath solution flow rates (0.5 and 3 ml/min). The neural activity at the highest flow rate is much larger and reflects the monotonously increasing relationship that saturates at 8 ml/min (*Figure 1—figure supplement 1*). In a small number of neurons (6 out of 110), increasing the flow rate produced a suppressive response (*Figure 1—figure supplement 1*). SGN terminals are unmyelinated (*Kim and Rutherford, 2016*; *Liberman, 1982*). We inferred that the SGN terminals could undergo minute displacement, subject to OCM (*Chen et al., 2011*; *Jawadi et al., 2016*; *Karavitaki and Mountain, 2007*), raising the possibility of a direct mechanical pathway affecting AN responses in addition to synaptic transmission. To stimulate the unmyelinated dendritic terminals in vitro, SGNs were cultured on a polydimethylsiloxane (PDMS) substrate (*Cheng et al., 2010*). We displaced a single dendrite by substrate indentation on this platform, using a fire-polished glass pipette driven by a piezoelectric actuator ($S_d$, inset *Figure 1b*). Dendrite-substrate displacement evoked membrane depolarization and APs at the recording patch electrode (R, inset *Figure 1b*). Soma-substrate or direct soma displacement was used for extended recordings and voltage(V)-clamp experiments. In both stimulation configurations, rectangular or ramp displacements evoked either a subthreshold membrane depolarization or APs, depending on the amplitude or ramp velocity (slew rate) (*Figure 1b, c*). When SGNs were interrogated with stepped mechanical displacement, 12 out of 32 neurons showed an increase in firing rate. In contrast, in some SGNs, including those with spontaneous activity (n = 10), an increase in mechanical displacement amplitude induced a suppressive response. In 10 out of 32 SGNs, interrogation with stepped mechanical displacement did not affect the AP firing rate (*Figure 1—figure supplement 2*).

From a holding potential ($V_h$) of –70 mV, displacement evoked current ($I_{MA}$) (*Figure 1d*). The $I_{MA}$ amplitude ranged from 100 to 700 pA (426 ± 85 pA; n = 95). The $I_{MA}$ shows a bi-exponential decay over time with a fast ($\tau_1$) and slow ($\tau_2$) time constant. For $I_{MA}$ elicited with a 1.12 µm displacement, $\tau_1$ and $\tau_2$ were 3.6 ± 2.6 and 24 ± 5.7 ms (n = 17) for apical neurons, and 2.1 ± 1.1 and 17.0 ± 4.8 ms (n = 15) for basal neurons (*Figure 1—figure supplement 3*), respectively. The displacement-response relationships of these apical and basal neurons (*Figure 1e*), expressed as a channel open probability ($P_o$), were fitted with a single Boltzmann function (black curves). The half-maximal activation displacements ($X_{0.5}$) were 0.42 ± 0.01 µm (n = 14) and 0.35 ± 0.01 µm (n = 14) for $I_{MA}$ from apical and basal neurons, respectively (*Figure 1e*). Varying $V_h$ and using a constant displacement (~0.4 µm) yielded $I_{MA}$ with a linear I-V relationship and a reversal potential ($E_{MA}$) ~0 mV (–1.4 ± 2.1 mV; n = 17), which is consistent with a non-selective cationic conductance (*Figure 1f*). The $I_{MA}$ in SGNs was sensitive to an externally applied MA channel blocker, GsMTx4 (*Bae et al., 2011*). Application of 1 µM GsMTx4 decreased the current amplitude by ~52% (52 ± 8%, n = 5; *Figure 1h* inset). The half-maximal inhibitory concentration ($IC_{50}$) obtained from the dose-response curve was 0.9 ± 0.2 µM (n = 4; *Figure 1h*). Additionally, 1 µM GsMTx4 completely abolished the dendrite displacement-evoked APs, which was partially reversible after washout (*Figure 1g*).

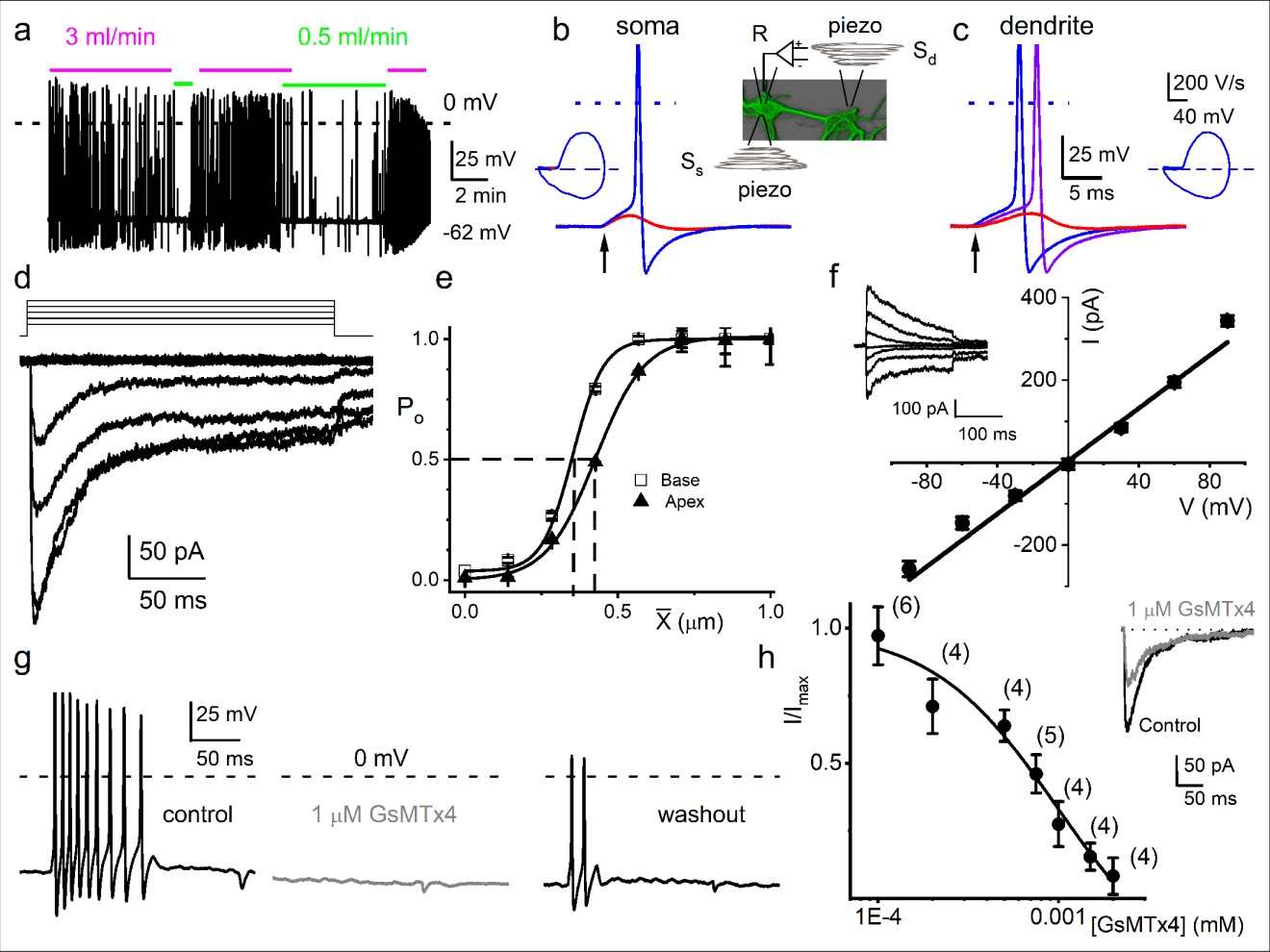

**Figure 1.** Mouse primary auditory neurons are mechanically sensitive. (**a**) Responses of 8-week-old basal spiral ganglion neuron (SGN) to bath solution flow rate shown with colored bars (3 ml/min, magenta; 0.5 ml/min, green). $V_{rest}$ = –62 mV. (**b**) Mechanical displacement (20 ms rectangular-pulse injection) of apical SGN soma, subthreshold (0.15 μm, red), and threshold (0.4 μm, blue) responses. Arrow indicates pulse initiation. Inset depicts the patch-clamp recording electrode (R) and stimulating probe (piezoelectric actuator represented as spring), placed ~180° from the patch electrode at the soma ($S_s$) or on the substrate to stimulate dendrites ($S_d$). (**c**) Responses of basal SGN to dendrite-substrate ramp-displacement at slew rates (μm/ms) 0.1 (red), 0.5 (purple), and 1.0 (blue). Arrow indicates the time of pulse initiation. Action potential (AP) latency increased as the slew rate decreased rate, 1 μm/ms, latency = 4 ± 0.1 ms; 0.5 μm/ms = 6 ± 0.2 ms (n = 14). The corresponding phase plots (dV/dt versus V) of the responses are shown next to the traces. (**d**) Current traces from displacement-clamp (X = 0–1.2 μm, step size ΔX = ~0.24 μm) recordings at a holding voltage of –70 mV. Data are obtained from apical SGNs; for basal SGNs data, see S2. (**e**) Summary data of displacement-response relationship of $I_{MA}$ represented as channel open probability ($P_o$) as a function of displacement, fitted by a single Boltzmann function. Data are from apical (▲) and basal (□) SGNs. The one-half-maximum displacements ($X_{0.5}$) are indicated by the vertical dashed lines ($X_{0.5}$ = 0.42 ± 0.01 μm (apical SGNs) and 0.35 ± 0.01 μm (basal SGNs), n = 14 for both). (**f**) Average I-V relationship of MA currents. Inset shows traces of $I_{MA}$ evoked at different holding voltages (–90 to 90 mV, ΔV = 30 mV, X = 0.4 μm). The regression line indicates a reversal potential of the MA current ($E_{MA}$) of –1.4 ± 2.1 mV (cumulative data from apical SGNs, n = 17). The whole-cell conductance derived from the regression line is 3.2 ± 0.2 nS (n = 17). (**g**) APs from basal SGN evoked in response to 0.8 μm displacement of neurite substrate (first black trace; $V_{rest}$ = –59 mV; dashed line = 0 mV). Effect of 1 μM GsMTx4 (middle gray trace) and partial recovery after washout (last black trace; n = 5). (**h**) Dose-response relationship of $I_{MA}$ as function of blocker GsMTx4, $IC_{50}$ = 0.9 ± 0.2 μM (n = 4). Numbers of SGNs at different concentrations of GsMTx4 are shown in brackets. Inset shows $I_{MA}$ in response to a 0.3 μm displacement at –70 mV holding voltage (gray trace is 1 μM GsMTx4; black trace is control).

The online version of this article includes the following source data and figure supplement(s) for figure 1:

**Source data 1.** Mechanical sensitivity of SGNs.

**Figure supplement 1.** Effect of bath solution flow rate on the spiral ganglion neuron (SGN) firing rate.

**Figure supplement 1—source data 1.** Effects of bath flow rate on SGN response properties.

**Figure supplement 2.** Varied responses to combined mechanical and electrical stimulation of spiral ganglion neurons (SGNs).

**Figure supplement 2—source data 1.** Combined current and mechanical stimulation of SGNs.

*Figure 1 continued on next page*

*Figure 1 continued*

**Figure supplement 3.** Displacement-clamp current recordings from 8-week-old adult spiral ganglion neurons (SGNs).

**Figure supplement 3—source data 1.** Displacement-clamp of SGNs.

**Figure supplement 4.** Chinchilla primary auditory neurons are equally mechanically sensitive.

**Figure supplement 4—source data 1.** Chinchilla auditory neurons are mechanically sensitive.

**Figure supplement 5.** Mechanical sensitivity of mouse vestibular ganglion neurons.

**Figure supplement 5—source data 1.** Mouse auditory neuron response properties to mechanical stimuli.

**Figure supplement 6.** Response properties of adult mouse spiral ganglion neurons (SGNs) to mechanical stimulation.

**Figure supplement 6—source data 1.** Mouse SGN responses to mechanical sinuisodal stimuli.

**Figure supplement 7.** Mouse spiral ganglion neuron (SGN) response properties to sinusoidal mechanical and current stimulation.

**Figure supplement 7—source data 1.** Combined current and mechanical sensitive of mouse SGNs.

## Response properties of chinchilla auditory and mouse vestibular neurons to stepped and sinusoidal mechanical stimulation

If the mechanosensory features of the SGNs are fundamental to auditory information coding, we would expect the phenomena to transcend species differences. Auditory neurons' mechanical sensitivity was not restricted to the mouse: adult chinchilla SGNs were similarly, but not identically, responsive to mechanical stimulation, validating the potential physiological relevance across species. Shown are exemplary APs from chinchilla SGNs in response to the soma's current injection and mechanical stimulation (*Figure 1—figure supplement 4*). The current elicited with mechanical displacement from –70 mV $V_h$, and the summary of the corresponding displacement-response relationship was also fitted with a single Boltzmann function with $X_{0.5}$ of 0.49 ± 0.05 (n = 9; *Figure 1—figure supplement 4*). The I-V relationship from chinchilla SGN $I_{MA}$ produced $E_{MA}$ = –8.1 ± 5.5 mV; n = 5 (*Figure 1—figure supplement 4*). The results suggest that $I_{MA}$ in chinchilla SGNs share features with those observed in the mouse.

A related issue is how specific the mechanical sensitivity of SGNs is relative to other mechanosensitive systems. Responses to mechanical displacement steps were also observed in vestibular neurons (VNs) (*Figure 1—figure supplement 5*). The displacement-response relationship was approximated using a two-state Boltzmann function (*Figure 1—figure supplement 5*). Compared to auditory neurons, mouse VNs were less responsive to stepped mechanical stimulation on average approximately threefold in magnitude. Additionally, responses to mechanical stimulation frequency >10 Hz were attenuated in VNs (*Figure 1—figure supplement 5*), compared with SGNs' responses in *Figure 1—figure supplement 7*. The magnitude of mechanical displacement required to elicit responses of VNs was comparable to those reported for dorsal root ganglion (DRG) neurons (*Finno et al., 2019*; *Viatchenko-Karpinski and Gu, 2016*).

SGNs show a variety of responses to current injection. Current-evoked responses from 5-week-old SGN range from fast to intermediate to slow adapting activity. Additionally, a fraction (5–10%) of SGNs are spontaneously active (*Adamson et al., 2002*; *Wang et al., 2013*). This diversity of responses is observed in adult apical and basal neurons (*Lv et al., 2012*). The question arises on how the response diversity to current injection relates to responsivity to mechanical stimulation. Apical and basal SGNs showed different response thresholds, with apical neurons showing higher sensitivity (*Figure 1—figure supplement 6*). The response latency and excitability of SGNs depended on stimulus type (step versus ramped pulse) and location (soma or dendritic substrate). For example, in fast-adapting SGNs, the first-spike latency increased from 4.0 ± 0.1 to 6.0 ± 0.2 ms (n = 14; p = 0.01) when the slew rate was reduced from 1 to 0.5 μm/ms; subsequent reduction in slew rate caused prolonged latency, subthreshold membrane depolarization, or failed responses (<0.001 μm/ms; *Figure 1—figure supplement 6*).

We used sinusoidal mechanical displacement as a proxy for in vitro sound stimulation at different frequencies, recording from mouse SGNs, and the responses were also variable (*Figure 1—figure supplement 7*). AP firing increased with increasing amplitude of mechanical stimulation (*Figure 1—figure supplement 7*) and was phase-locked. In some SGNs (4 out of 11), the responses were attenuated with larger mechanical displacement. A train of varying frequencies of mechanical stimulation, in decade steps, shows the SGNs responded to and are phase-locked to specific frequencies up to

100 Hz, but responses were absent at 1000 Hz. In contrast, the response properties of SGNs reached 1000 Hz sinusoidal current injection. Responses to current injection were broader than mechanical stimulation (*Figure 1*, *Figure 1—figure supplement 7*).

## Amplitude and phase of combined current and mechanical sinusoidal stimulation affect response rate and timing

Because sound waves are converted into mechanical vibrations transmitted via the middle ear to the cochlea, which converts them into neural signals, the question arises whether the mechanical responses shown here play a role in hearing. As AN fibers traverse different compartments of the OC (osseous spiral lamina, basilar membrane) to innervate IHC, they lose their myelin sheath at the habenula perforata (*Morrison et al., 1975*; *Figure 2a*). The unmyelinated terminal is subject to sound-evoked displacement or pressure changes. Direct examination of potential synergistic or antagonistic effects between neurotransmission and mechanical signaling at or near the AN synaptic terminal is currently not technically possible; however, the interaction of their proxies of current injection and substrate vibration can be examined. We applied sinusoidal currents and mechanical displacements to produce more physiological stimuli that mimic sound-evoked neurotransmission and movement (see Materials and methods).

We find that the two stimulus modalities interact: mechanical and current stimuli, which are subthreshold and can be suprathreshold when combined (*Figure 2b*). Since the exact relationship in amplitude and phase between synaptic events and the mechanical displacement or deformation hypothesized to affect the afferent dendrite is unknown, we explored different amplitude (*Figure 2*) and phase (*Figure 3*) relationships between current and mechanical stimulation. First, the interaction was determined using in-phase stimulation: SGNs were primed with sustained subthreshold current, and the displacement-response relationship was tested. The mechanical responsiveness increased with increased current injection, shifting the rate curves to lower displacement values (*Figure 2c*). Increasing subthreshold mechanical displacement moved the rate curves to lower current values (*Figure 2d*). The $X_{0.5}$ derived from *Figure 2c* as a current priming function has a linear relationship, with a slope of –1.4 µm/nA (*Figure 2e*). As a function of priming mechanical displacement, the converse half-maximum current (I0.5; *Figure 2d*) also shows a linear relationship, with a slope of –0.5 nA/µm (*Figure 2f*). These orderly interactions between intrinsic mechanical responsiveness and electrical activity suggest that the increase in mechanical displacement with increasing sound intensity could have a monotonic effect on the spike rate.

The effect of combined stimulation on phase locking was studied with in-phase sinusoidal current and mechanical displacement overlapped for ~2 s. *Figure 2g* (upper panel) shows a slowly adapting SGN response to current stimulation (6-week-old basal SGN) with a firing rate of 36 ± 5 spikes/s, which increased to 50 ± 3 spikes/s (n = 9; p = 0.01) upon paired mechanical stimulation applied to the soma. Synchronization between stimulus and response was measured with vector strength (VS) (*Goldberg and Brown, 1969*): VS to the 100 Hz current injection alone was 0.82 ± 0.05 and increased with paired stimulation to 0.98 ± 0.01 (n = 9; p = 0.01). For the moderately adapting SGN (*Figure 2g* lower panel; 5-week-old apical SGN), the VS increased from 0.72 ± 0.06 to 0.82 ± 0.07 (n = 8; p = 0.04) after paired stimulation. The converse paradigm, where mechanical stimulation preceded combined current and mechanical stimulation, is illustrated for two fast-adapting SGNs. They showed a significantly increased AP firing rate when the two stimuli overlapped (*Figure 2h*), with high VS values (0.94 and 0.9). For this set of experiments (28 SGNs) in which the pre-paired stimulation yielded non-zero VS (see Materials and methods), 21 SGNs (75%) showed a significant increase in VS (p < 0.05). In the remaining 7 SGNs (25%), the VS was reduced (*Figure 2i–j*, shown in green symbol and line). The summary data show that combined current and mechanical stimulation tended to alter the VS, suggesting the two stimuli interact to shape the response properties of SGNs (*Figure 2i–j*; *Table 1*). When the response to mechanical stimulation was reduced by application of GsMTx4 (1 µM), dual current and displacement-responses resulted in a reduced VS (*Figure 2—figure supplement 1*, *Table 2*).

Because it is plausible that the amplitude and the phase of displacement of the IHC stereociliary bundle vary relative to the mechanical events affecting SGN dendrites, depending on sound parameters, the effects of sinusoidal displacement, and current injection at different phase angles and amplitudes were tested. The top row of *Figure 3a–c* shows SGN responses to combined mechanical (magenta) and current (green) stimulation – the 50 Hz stimuli are in-phase but are changed in

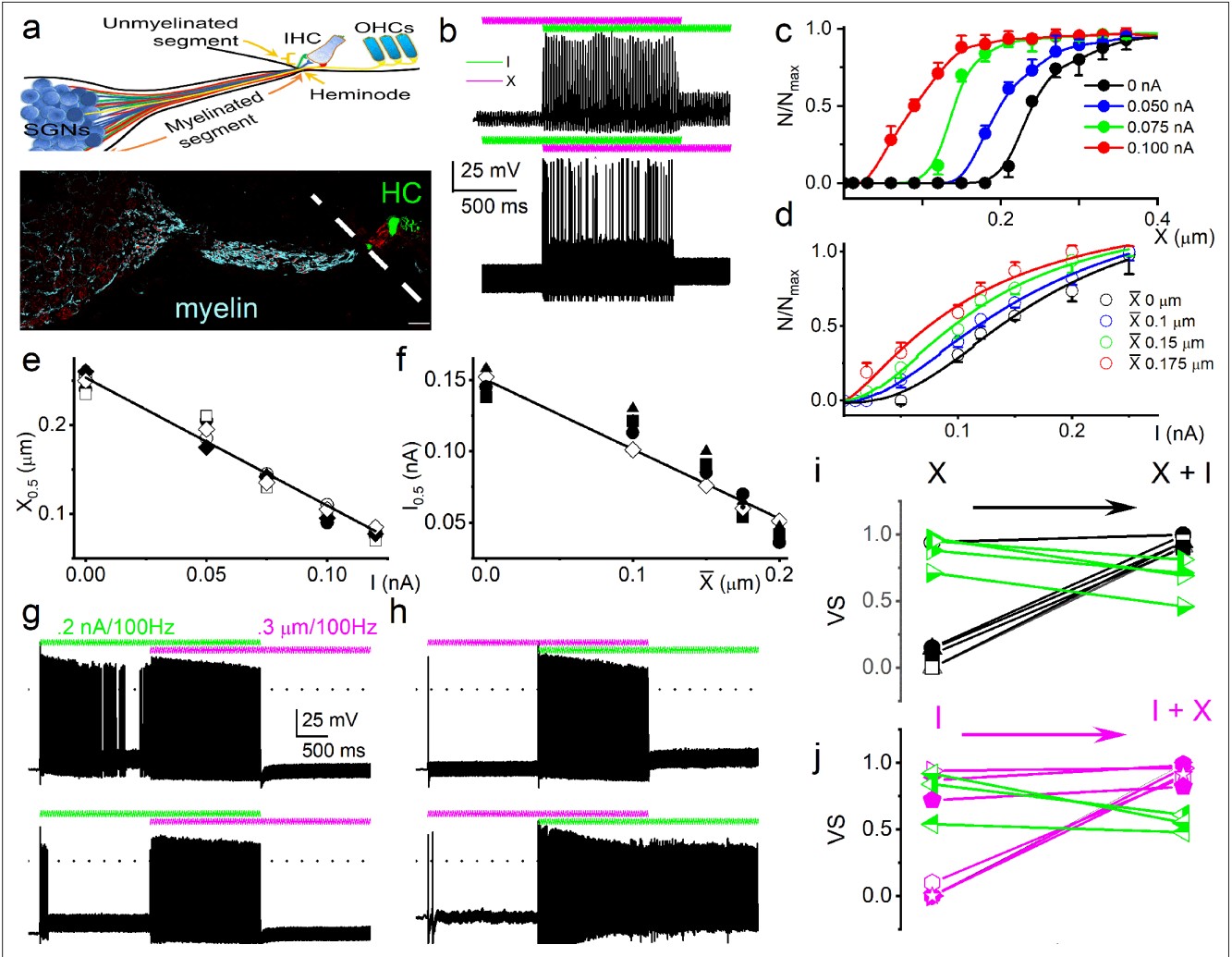

**Figure 2.** Mechanical stimulation of spiral ganglion neuron (SGN) cell body increased sensitivity to current injection and affected phase locking. (**a**) (Upper panel) Shown is a schematic diagram of the SGN and projections to inner hair cells (IHCs) and outer hair cells (OHCs). The myelinated and unmyelinated segments and nerve fibers are indicated. The heminode, the action potential (AP) initiation site, is labeled. (Lower panel) Horizontal section of whole-mount intact SGNs and HCs of a 5-week-old mouse cochlea, showing neuron (in red, anti-TUJ1) and myelin (in cyan, anti-myelin basic protein). The white dashed line shows the expected location of the basilar membrane. HCs are in green (anti-Myo7a). Note that the nerve terminals are devoid of myelin. Scale bar = 20 μm. (**b**) (Upper panel) Voltage response to combined subthreshold sinusoidal mechanical displacement (in magenta; 50 Hz, 0.1 μm, ~ 2 s) and subthreshold sinusoidal current injection (in green, 50 Hz, 0.05-nA, ~ 2 s), delivered in-phase. Current injection and mechanical displacement overlapped for ~1 s. Combined subthreshold mechanical and current stimulation evoked APs. (Lower panel) We show conditions similar to the upper panel, but subthreshold current injection preceded subthreshold mechanical displacement here. Subthreshold stimuli, when paired, became suprathreshold. (**c–d**) Spike rate, normalized to maximum, as a function of displacement (**c**), and current injection (**d**). The leading stimulus was primed with a subthreshold sinusoidal current or displacement, respectively. (**e–f**) The extent of sensitivity to current and displacement is represented as a plot of the half-maximum displacement ($X_{0.5}$) and half-maximum current ($I_{0.5}$) as a function of current and displacement. The slopes of the corresponding linear plots were, –1.4 μm/nA (n = 5) (**e**) and –0.5 nA/μm (n = 5) (**f**). (**g**) (Upper panel) Slowly adapting SGN response to 0.2 nA injection at 100 Hz, overlapping with a pre-determined threshold (0.3 μm displacement at 100 Hz) mechanical stimulus. For this example, the vector strength (VS) transitioned from 0.82 to 0.98 upon combined (current plus mechanical displacement) stimulation (see *Table 1* for summary data). (Lower panel) Moderately adapting SGN, stimulated with the same stimuli as the upper panel, transitioned from VS of 0.72–0.82 (*Table 1*). (**h**) (Upper and lower panels) Stimulation of fast-adapting SGN in response to 100 Hz, a 0.3 μm displacement superimposed with 100 Hz 0.2 nA current injection. For the examples shown, the VS transitioned from unmeasurable to 0.94 for the upper panel and 0.9 for the lower panel. (**i–j**) A summary of VS changes when either current or displacement is used as a primer for concurrent stimulation (*Figure 1*, *Table 1*, *Figure 1—figure supplement 7*).

The online version of this article includes the following source data and figure supplement(s) for figure 2:

**Source data 1.** SGN mechanical responses.

**Figure supplement 1.** GsMTx4 reduced facilitation and spike timing after dual current injection and mechanical stimulation.

**Figure supplement 1—source data 1.** Combined mechanical and current stimuli in SGN and GsMTx4 effects.

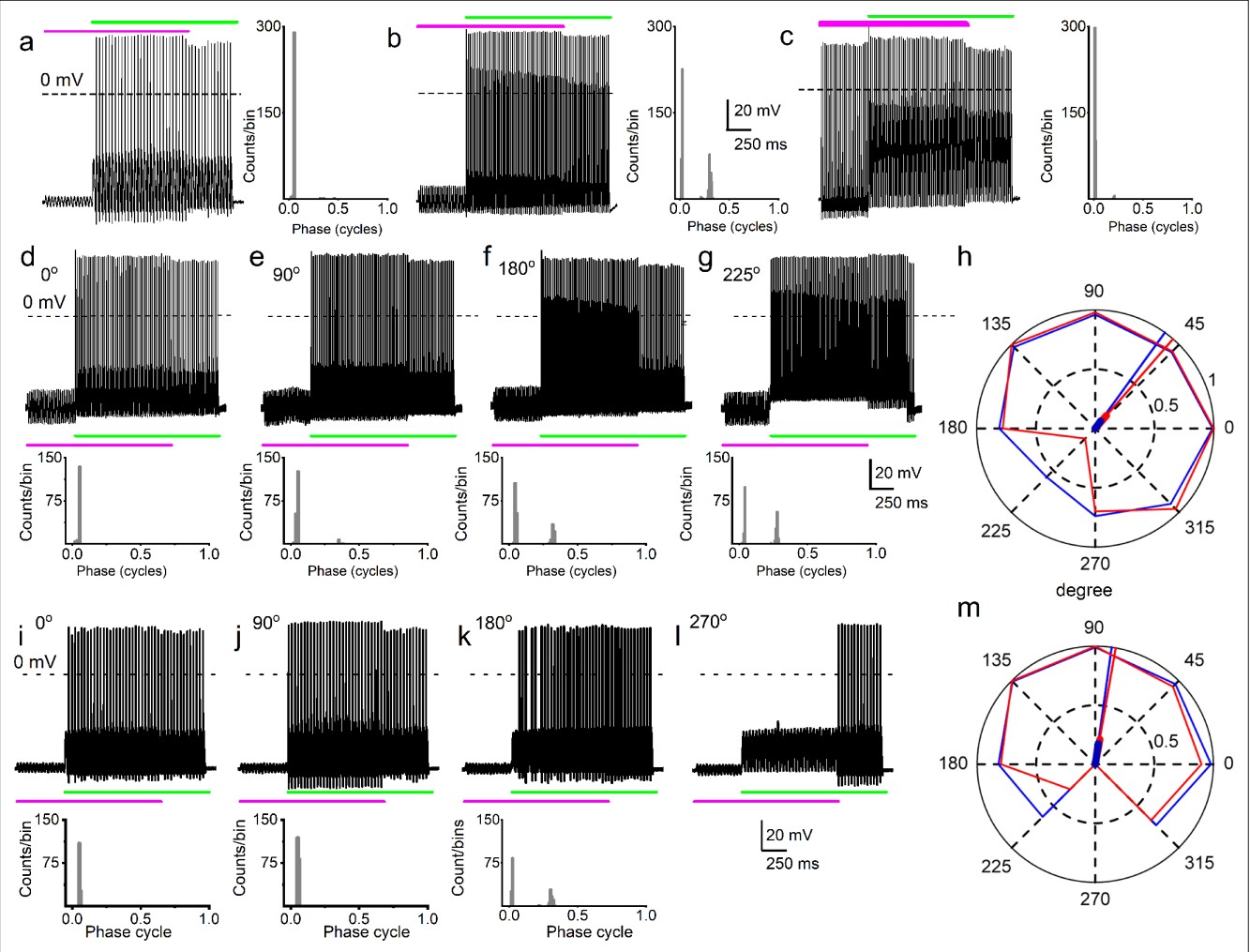

**Figure 3.** Simultaneous mechanical and electrical stimulation of spiral ganglion neurons (SGNs) can generate peak-splitting and rate suppression. (**a**) SGN response properties and corresponding cycle histograms generated by simultaneous sinusoidal mechanical subthreshold displacement (0.1 µm, 50 Hz; magenta) and sinusoidal current injection (0.2 nA, 50 Hz; green). The relative phase angle of the two stimuli was at 0°. The amplitude criterion of a valid action potential (AP) was 0 mV (dashed line). (**b, c**) The cycle histograms transitioned to two peaks and back to one peak as the amplitude of mechanical displacement increased to 0.2 µm (**b**) and 0.3 µm (**c**). (**d–g**) Response properties and corresponding cycle histograms for the same SGN as in (**a–c**), but using different relative phase angles (0°, 90°, 180°, 225°) between mechanical displacement (0.15 µm) and current injection (0.3 nA). At 180° and 225° phase angles, two peaks are evident in the cycle histograms shown in the panels below. (**h**) Polar plot summarizing data averaged over three stimulus repetitions derived from the SGN illustrated in d–g. The angular scale specifies the phase lead of the current relative to the mechanical stimulus and the two data points at each of the eight angles plotted are derived from data as shown in individual panels d–g: average rate is shown in blue and is normalized to its maximum (listed in *Table 3*); VS is shown in red. The data are summarized with a vector (resultant of vectorial addition of data at eight angles) whose magnitude and angle are shown with thick and thin solid lines, respectively, using the same color code. The close angular alignment of the red and blue lines shows that a high firing rate is accompanied by strong phase locking, and vice versa that peak-splitting is associated with low firing rates. (**i–l**) Phase angle-dependent response reduction (180°), and suppression (270°) during combined stimulation for a different neuron. (**m**) A polar plot as in **h** but for the neuron illustrated in (**i–l**). Data for the polar plots spike rates, magnitude and phase of resultant vectors, and statistics are provided in *Table 3*.

The online version of this article includes the following source data and figure supplement(s) for figure 3:

**Source data 1.** Combined current and mechanical stimuli and SGN peak splitting rate, and suppression.

**Figure supplement 1.** Combined displacement and current injection yield more than two action potentials per cycle in chinchilla spiral ganglion neurons (SGNs).

**Figure supplement 1—source data 1.** Combined current and mechanical stimulation of SGN.

**Figure supplement 2.** Mechanical sensitivity of chinchilla spiral ganglion neurons (SGNs).

**Figure supplement 2—source data 1.** Mechanical sensitivity of Chinchilla SGN.

*Figure 3 continued on next page*

*Figure 3 continued*

**Figure supplement 3.** Additional examples of the relationship between firing rate and vector strength (VS) for different relative phase angles between mechanical and current stimulation.

relative amplitude by increasing the displacement amplitude in steps of 0.1 µm. In this figure, the response as a function of post-stimulus time is shown as well as a cycle histogram (firing as a function of stimulus phase) to its right (*Figure 3a–c*) or below (*Figure 3d–g and i–k*). Combining subthreshold (0.1 µm) mechanical stimulation with a suprathreshold (0.2 nA) current triggered a highly phase-locked response, as illustrated by the cycle histogram (*Figure 3a*, right panel), which shows that the vast majority of spikes occurred over a very narrow range of phase angles. An increase of the mechanical stimulus amplitude to 0.2 µm causes spikes to appear at a different phase, giving rise to a second peak in the cycle histogram, reminiscent of the in vivo phenomenon of peak-splitting (*Kiang and Moxon, 1972*; *Liberman and Kiang, 1984*). With a further increase in amplitude to 0.3 µm, the cycle histogram is again unimodal (*Figure 3c*). Peak-splitting with paired mechanical-current stimulation was also observed in chinchilla SGNs (*Figure 3—figure supplements 1–2*). On rare occasions (2 out of 48 SGNs), 3 APs were evoked per cycle, as shown for a chinchilla SGN (*Figure 3—figure supplement 1*). To test the effect of the relative phase of the mechanical stimulus, it was changed in 45° increments with respect to the current while measuring response rate and phase, illustrated for two neurons in *Figure 3d–h and i–m*. Peak-splitting could be observed at some phase angles (*Figure 3f, g and k*) and not at others (*Figure 3d, i and j*, *Figure 3—figure supplement 1*). Phase angles between mechanical and current stimuli that resulted in peak-splitting also resulted in lower spike rates. We illustrated and quantified with polar graphs showing VS (red) and spike rate (blue, normalized to the maximum rate for each fiber) for each increasing phase lead of the current stimulus relative to the mechanical stimulation (*Figure 3h and m*, *Figure 3—figure supplement 2*). Because a drop in VS necessarily accompanies peak-splitting, the latter's occurrence is reflected in reduced VS values. For example, for in-phase current and mechanical stimulation (*Figure 3d*, 0°), high firing rate, and VS are obtained, resulting in values near 1 (*Figure 3h*: blue and red lines touch outer, solid circle of magnitude 1). However, firing rate and VS are lower for angles where peak-splitting is observed, for example, at 180° (*Figure 3f*) and particularly 225° (*Figure 3g*): for these phase angles, the polar plot shows lower values of both rate (blue) and VS (red). To quantify this trend, we computed a normalized resultant vector for each polar plot, which summarizes the trend of rate and VS when the relative phase between current and mechanical stimulation is varied. The magnitude and angle of the resultant vector are indicated with thick and thin solid lines, respectively (*Figure 3h and m*, see *Table 3* for values and statistics). In stark resemblance to observations in vivo, there was a decrease in spike rate at the phase angles where peak-splitting occurred, illustrated by the alignment in resultant vectors for rate and VS (*Figure 3h and m*, *Figure 3—figure supplement 2*: red and blue lines). In vivo, this phenomenon has been called 'Nelson's notch' (*Kiang and Moxon, 1972*; *Liberman and Kiang, 1984*), which had not been recapitulated in vitro. This decrease could result in spike rates lower than those elicited by the single-stimulus conditions, for example, in *Figure 3l*, the spike rate during overlap is lower than due to current injection only. Thus, paired stimuli can not only generate a monotonic increase in spike rate (*Figure 2c–f*) but can also cause response suppression. Further examples of the correlation between spike rate and VS are shown in *Figure 3—figure supplement 3* from series of measurements with relative phase between current and mechanical stimuli as the independent variable.

## AN responses in the absence of synaptic transmission

To examine whether AN fibers are influenced directly by mechanical displacement in vivo, we administered synaptic transmission blockers through the cochlear round window (RW) while recording from single fibers. Reports have shown two response regions in frequency tuning curves: a 'tip' of sharp tuning observed at low sound levels, and a 'tail' of coarse tuning (usually below the CF) at high sound levels (*Kiang et al., 1986*). These two regions are differentially vulnerable to experimental manipulations (*Kiang et al., 1986*). The inherent mechanical sensitivity of AN fibers demonstrated in vitro is observed for substrate motion at sub-micrometer displacements (<100 nm, e.g., *Figure 2c*). But note that temporal effects can occur at subthreshold displacements, that is, that do not trigger spikes themselves (e.g., *Figure 3*). Cochlear displacements of this magnitude have been measured in vivo, particularly toward the cochlear apex and at high sound pressure levels (SPLs), where frequency

**Table 1.** Summary of changes in spike rate I and VS using in-phase current and mechanical stimulations.

The significance level of VS is given in the column with p-values obtained with the Rayleigh test for uniformity (*Mardia, 1972*).

| SGN | Current (I) R (spike/s) VS | | Rayleigh p | Mech stim (X) R (spike/s) VS | | Rayleigh p | I + X R (spike/s) VS | | Rayleigh p |
|---|---|---|---|---|---|---|---|---|---|
| 13830002 | 0.5 | NA | NA | 0 | NA | NA | 40 | 0.96 | 0.0013 |
| 13830003 | 0 | NA | NA | 0 | NA | NA | 40 | 0.91 | 0.003 |
| 13903009 | 50 | 0.97 | 0.01 | 0 | NA | NA | 74 | 0.95 | 0.001 |
| 13903030 | 7.5 | 0.98 | 0.02 | 2.5 | 0.14 | 0.11 | 95 | 0.94 | 0.011 |
| 13903043 | 21 | 0.96 | 0.01 | 14 | 0.95 | 0.02 | 99 | 0.99 | 0.0003 |
| 13903044 | 11.5 | 0.95 | 0.01 | 51 | 0.98 | 0.01 | 100 | 0.99 | 0.0086 |
| 13903029 | 23 | 0.94 | 0.04 | 0 | NA | NA | 89 | 0.96 | 0.0002 |
| 13903010 | 49 | 0.98 | 0.01 | 0.5 | NA | NA | 69 | 0.94 | 0.0012 |
| 13903058 | 45 | 0.98 | 0.01 | 12 | 0.97 | 0.02 | 96 | 0.97 | 0.0003 |
| 13903059 | 24.5 | 0.94 | 0.03 | 50.5 | 0.98 | 0.01 | 92 | 0.96 | 0.002 |
| 13904007 | 0.5 | NA | NA | 0 | NA | NA | 99.5 | 0.99 | 0.009 |
| 13904008 | 0 | NA | NA | 22.5 | 0.94 | 0.03 | 100 | 0.997 | 0.0001 |
| 13904023 | 1.5 | 0.15 | 0.43 | 0 | NA | NA | 99.5 | 0.99 | 0.0002 |
| 13904024 | 0 | NA | NA | 1 | NA | NA | 100 | 0.96 | 0.003 |
| 13904037 | 2 | 0.1 | 0.67 | 0 | NA | NA | 55 | 0.89 | 0.008 |
| 13904038 | 0 | NA | NA | 2.5 | 0.15 | 0.32 | 50.5 | 0.995 | 0.008 |
| 13909008 | 0.5 | NA | NA | 0 | NA | NA | 64 | 0.96 | 0.012 |
| 13909009 | 0 | NA | NA | 1.5 | 0.12 | 0.52 | 62 | 0.96 | 0.013 |
| 13909024 | 36 | 0.87 | 0.02 | 0 | NA | NA | 50 | 0.98 | 0.011 |
| 13909025 | 0 | NA | NA | 0.5 | NA | NA | 50.5 | 0.95 | 0.012 |
| 13909040 | 0 | NA | NA | 0 | NA | NA | 16 | 0.94 | 0.007 |
| 13909041 | 0 | NA | NA | 0 | NA | NA | 13 | 0.75 | 0.052 |
| 13913008* | 45.5 | 0.72 | 0.04 | 0 | NA | NA | 43 | 0.82 | 0.036 |
| 13913009* | 35 | 0.87 | 0.03 | 1 | NA | NA | 50.5 | 0.9 | 0.027 |
| 14900012* | 18 | 0.76 | 0.02 | 3 | 0.15 | 0.34 | 64.5 | 0.95 | 0.011 |
| 14903018* | 0.5 | NA | NA | 5 | 0.2 | 0.36 | 31.5 | 0.87 | 0.021 |
| 14903021* | 1.5 | 0.1 | 0.25 | 0.5 | NA | NA | 45.5 | 0.91 | 0.010 |
| 14903083 | 32.5 | 0.68 | 0.02 | 0.5 | NA | NA | 44.5 | 0.83 | 0.008 |
| 15800101 | 0.5 | NA | NA | 3 | 0.1 | 0.31 | 72.5 | 0.96 | 0.006 |
| 15800110 | 56 | 0.95 | 0.01 | 1 | NA | NA | 62 | 0.93 | 0.007 |
| 15800099 | 32 | 0.94 | 0.02 | 0 | NA | NA | 70.5 | 0.96 | 0.041 |
| 15810007 | 56 | 0.98 | 0.01 | 3.5 | 0.2 | 0.14 | 58 | 0.97 | 0.020 |
| 15800291 | 0 | NA | NA | 5 | 0.25 | 0.08 | 94.5 | 0.99 | 0.001 |
| 15800292 | 0.5 | NA | NA | 3 | 0.10 | 0.56 | 56.5 | 0.92 | 0.005 |
| 15800293 | 3 | 0.15 | 0.09 | 1 | NA | NA | 95.5 | 0.99 | 0.0001 |

*Table 1 continued on next page*

*Table 1 continued*

| SGN | Current (I) R (spike/s) VS | | Rayleigh p | Mech stim (X) R (spike/s) VS | | Rayleigh p | I + X R (spike/s) VS | | Rayleigh p |
|---|---|---|---|---|---|---|---|---|---|
| 15800295 | 2.5 | 0.1 | 0.27 | 3 | 0.1 | 0.53 | 68.5 | 0.89 | 0.004 |
| 15800301 | 1 | NA | NA | 8 | 0.3 | 0.18 | 46.5 | 0.96 | 0.0002 |
| 15800308 | 48 | 0.87 | 0.02 | 1 | NA | NA | 92 | 0.98 | 0.003 |
| 15800319 | 1 | NA | NA | 7 | 0.25 | 0.11 | 89.5 | 0.94 | 0.003 |
| 15800320 | 0 | NA | NA | 5 | 0.3 | 0.11 | 49.5 | 0.91 | 0.003 |
| 15800401 | 16.5 | 0.71 | 0.02 | 17 | 0.55 | 0.03 | 56.5 | 0.89 | 0.021 |
| 15800402 | 7 | 0.25 | 0.05 | 1 | NA | NA | 42.5 | 0.91 | 0.025 |
| 15800511 | 3 | 0.1 | 0.18 | 1 | NA | NA | 40 | 0.94 | 0.008 |
| 15800513 | 5 | 0.2 | 0.09 | 6 | 0.14 | 0.17 | 52 | 0.94 | 0.006 |
| 15800515 | 1 | NA | NA | 94.5 | 0.98 | 0.003 | 89.5 | 0.99 | 0.002 |
| 16900101 | 0 | NA | NA | 97.5 | 0.95 | 0.01 | 99.5 | 0.998 | 0.002 |
| 16900107 | 5 | 0.2 | 0.05 | 3 | 0.1 | 0.21 | 75.5 | 0.96 | 0.001 |
| 16900111 | 9 | 0.3 | 0.05 | 5 | 0.15 | 0.11 | 98 | 0.995 | 0.007 |
| 16900123 | 1 | NA | NA | 6 | 0.12 | 0.24 | 72.5 | 0.96 | 0.0002 |
| 16900127 | 19 | 0.74 | 0.01 | 1 | NA | NA | 92 | 0.95 | 0.008 |
| 16900131 | 9 | 0.25 | 0.03 | 1 | NA | NA | 45.5 | 0.75 | 0.037 |
| 16900139 | 5 | 0.2 | 0.12 | 4 | 0.1 | 0.67 | 56 | 0.9 | 0.009 |

SGN = spiral ganglion neuron, R (spike rate) = spike/s, VS = vector strength, * substrate dendrite displacement, NA = not applicable.

tuning is poor (*Cooper and Rhode, 1995*; *Lee et al., 2015*; *Lee et al., 2016a*). The in vitro results show interactive effects of current injection and mechanical stimulation (*Figures 2 and 3*), so it is difficult to predict the response that would remain when one modality (synaptically evoked spiking) is abolished. Nevertheless, the expectation is that in vivo, in the presence of synaptic blockers, some degree of response will persist at high SPLs, both in normal and in traumatized cochleae. In contrast, if direct mechanical effects on AN dendrites have no role in responses to acoustic stimuli, synaptic blockers are expected to weaken all responses of a single fiber, irrespective of stimulus frequency.

We first characterized frequency tuning in single AN fibers during one or more electrode penetrations through the AN, recorded at the internal auditory meatus. After a sample of frequency tuning curves was obtained (*Figure 4a*), sufficient to record a frequency tuning profile as a function of recording depth, NBQX was applied through the RW. NBQX is a competitive antagonist of the ionotropic glutamate receptor, which blocks HC-AN synaptic transmission (*Grant et al., 2010*). The AN recording was then resumed with the same micropipette, left in situ during drug injection. NBQX application progressively

**Table 2.** Effects of GsMTx4 on spike rate (R) and VS.

The significance level of VS is given in the column with p-values obtained with the Rayleigh test for uniformity on the effects of GsMTx4.

| SGN | Current (I) R (spike/s) VS | | Mech stim (X) R (spike/s) VS | | I + X R (spike/s) VS | |
|---|---|---|---|---|---|---|
| 19830002 | 3 | 0.15 | 1.5 | 0.1 | 41 | 0.90 |
| GsMTx4 | 3 | 0.15 | 0 | NA | 5 | 0.24 |
| 19830002 | 45.5 | 0.74 | 1 | NA | 48 | 0.86 |
| GsMTx4 | 49.5 | 0.75 | 0 | NA | 42 | 0.72 |
| 19830043 | 9 | 0.62 | 5 | 0.3 | 31 | 0.89 |
| GsMTx4 | 11 | 0.62 | 0 | NA | 13 | 0.58 |
| 19830059 | 7 | 0.38 | 14 | 0.64 | 45 | 0.89 |
| GsMTx4 | 5 | 0.37 | 0 | NA | 11 | 0.37 |

SGN = spiral ganglion neuron, R (spike rate) = spike/s, VS = vector strength, NA = not applicable.

**Table 3.** Summary data for polar plots of *Figure 3* and *Figure 3—figure supplement 3*.
The header of the vertical columns provides a reference to the relevant figure panel. The top four datalines are for the polar plots based on spike rate. First line gives the maximum number of spikes, to which the rate plot was normalized. The second and third lines give the angle and magnitude of the resultant vector. Last line indicates level of significance (Rayleigh test of uniformity). The bottom three datalines give corresponding values based on measurement of phase locking.

| | Figure 3m | Figure 3—figure supplement 3a | Figure 3—figure supplement 3b | Figure 3—figure supplement 3c |
|---|---|---|---|---|
| Polar plot spike rate | | | | |
| Max #spikes | 147 | 150 | 210 | 278 | 150 |
| Angle resultant | 49.1 | 79.6 | 352.6 | 69.1 | 297.4 |
| Magnitude resultant | 0.15 | 0.21 | 0.19 | 0.23 | 0.22 |
| p-Value Rayleigh test | <0.001 | <0.001 | <0.001 | <0.001 | <0.001 |
| Polar plot phase locking | | | | |
| Angle resultant | 54.1 | 81.5 | 3.0 | 81.9 | 307.0 |
| Magnitude resultant | 0.08 | 0.18 | 0.12 | 0.17 | 0.20 |
| p-Value Rayleigh test | 0.003 | <0.001 | <0.001 | <0.001 | <0.001 |

decreased spontaneous- (*Figure 4d*) and sound-evoked (*Figure 4b*) spiking activity. Because these recordings are inherently blind, AN fibers without spontaneous activity can be detected only using brief high-intensity sound 'search' stimuli (see Materials and methods). *Figure 4a–b* shows tuning curves recorded before and after NBQX administration. Notably, after NBQX application and at low recording depths, fibers were excitable only at high SPLs. Tuning in these fibers was broad and shallow, with the lowest thresholds at low frequencies (*Figure 4c*). We surmised that these responses persisted due to the AN's mechanical sensitivity. Interpretation of the response is hampered by the difficulty of knowing to what frequency range the fibers were tuned before applying the blocker. Because the high-threshold, coarsely tuned responses were observed at recording depths where, pre-NBQX, neurons were tuned to high frequencies, it can be reasonably inferred that these responses origi- nated from fibers innervating the cochlear base. Fibers recorded at depths >250 µm were tuned to a restricted frequency range near 0.5–1 kHz, both pre- and post-drug applications (*Figure 4c*). Limited intra-cochlear diffusion of the drug likely accounts for lesser drug effects at these CFs (*Sadreev et al., 2019*; *Verschooten et al., 2015*).

While the data in *Figure 4a and b* are consistent with previous reports showing differences in the vulnerability of low- and high-threshold components of AN threshold curves, two fibers shown in *Figure 4e and f* provide a more direct indication that high-threshold, bowl-shaped tuning originated from basally located AN fibers. In these two fibers, the 'tip' but not the 'tail' region of the tuning curve disappeared during repeated measurement, such that only a bowl-shaped high-threshold response to low-frequency tones remained. For reference, *Figure 4e and f* shows the tuning curve of a fiber with CF of 3.4 kHz (blue, spontaneous rate of 8.7 spikes/s) recorded at 519 µm depth before NBQX application; the other curves (*Figure 4e and f*, red) were obtained post-NBQX. The dashed red curve in *Figure 4e* is the tuning curve of a fiber recorded at a similar depth (508 µm) after the NBQX appli- cation. It reveals a tip at 3.2 kHz (spontaneous rate of 15.3 spikes/s) with an elevated threshold and little separation between the threshold at the tip and tail. Presumably, the tuning curve before drug application was similar in shape to that of the fiber recorded at this depth pre-NBQX (blue curve). A second, subsequent measurement of the same fiber (solid red line, *Figure 4e*) shows a curve that is consistent with the initial tail but with the tip further attenuated. A similar progression was measured for a second fiber (*Figure 4f*), measured at a greater depth (652 µm, initial spontaneous rate = 23.3 spikes/s), and it showed a similar pattern of a tip near 3 kHz, which subsequently disappeared with a broad bowl-shaped curve remaining in a follow-up measurement (spontaneous rate = 15.9 spike/s). Combined, these data (*Figure 4a–f*) reveal a difference in vulnerability between the nerve fiber's tip

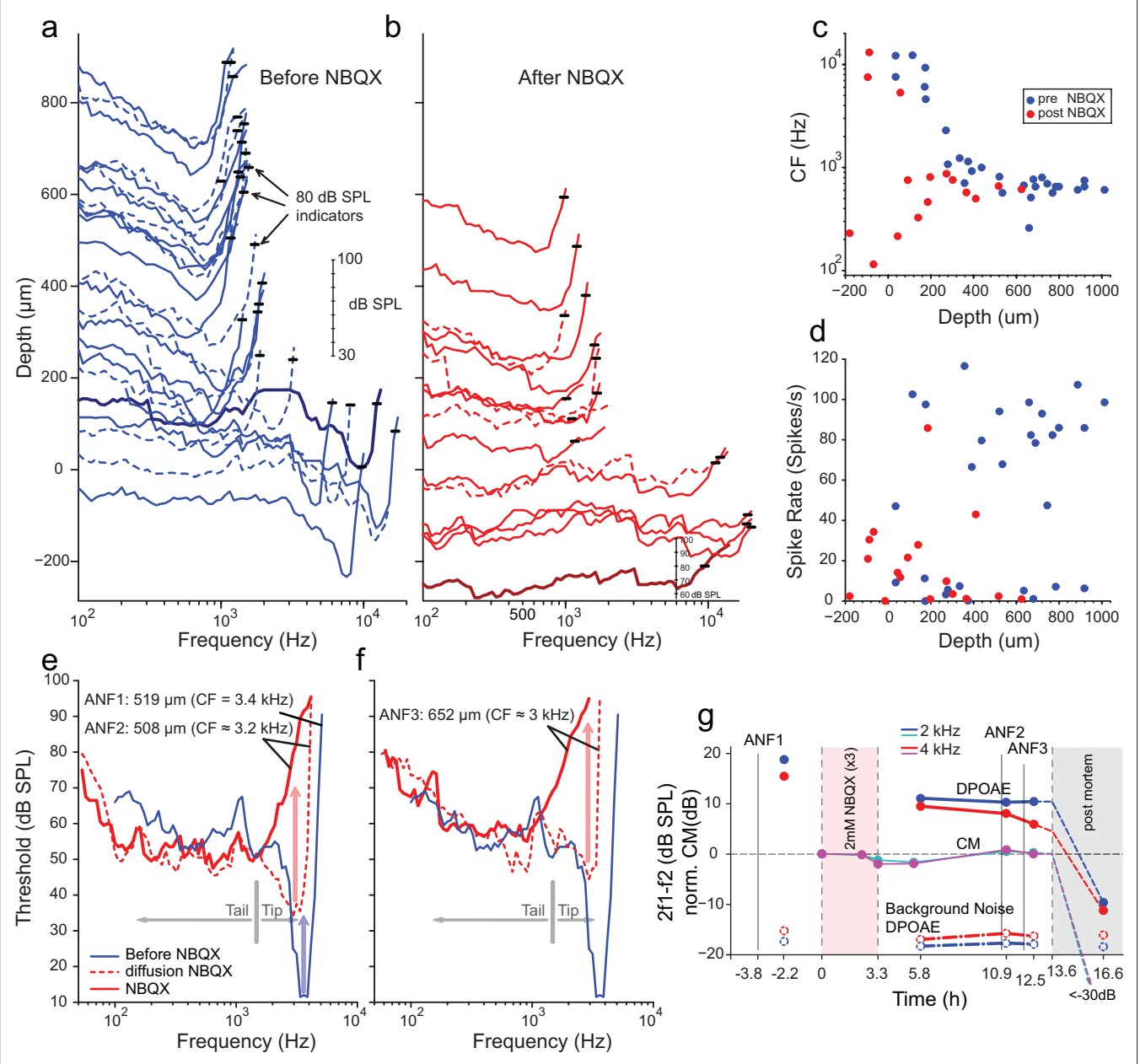

**Figure 4.** Effects of blocking synaptic transmission on auditory nerve (AN) activity in vivo. (**a–b**) Frequency threshold tuning curves of AN fibers from a single chinchilla, obtained at different recording depths in the nerve, before (blue) and after (red) round window application of 2,3-dihydroxy-6-nitro-7-sulfamoyl-benzo[f]quinoxaline (NBQX). Each curve shows the threshold (in dB sound pressure level [SPL]) of a single AN fiber over a range of frequencies: the curves are anchored to the y-axis (recording depth) by the small black horizontal bar at 80 dB SPL. The dB scale in **a** applies to all traces in (**a,b**). Dashed lines are sometimes used to disambiguate traces. After applying NBQX, frequency tuning vanishes at the shallow recording depths (toward the bottom of **a** and **b**), where high-frequency fibers were found before applying NBQX. Fibers tuned to lower frequencies, found at greater depth (toward the top of **a** and **b**), are less affected, likely due to limited diffusion of NBQX to the apical turns. (**c**) Characteristic frequencies (CFs) as a function of recording depth (frequency of lowest threshold) before and after application of NBQX in the same cochlea. In the initial recording sessions, before the application of NBQX (blue symbols), high CFs dominate at shallow recording depths, while at depths >250 μm CFs are between 0.5 and 1 kHz. After the application of NBQX (red symbols), the lowest thresholds of superficial fibers are predominantly at low frequencies (<1 kHz). (**d**) Spontaneous rates as a function of recording depth. Formatting as in c. (**e**) The effect of NBQX on repeated measurement of a threshold tuning curve of AN fiber 2 (ANF2) recorded initially (red dashed: CF = 3.2 kHz) and after several minutes (red solid: CF = 630 Hz), compared to a tuning curve of a fiber (ANF1) with comparable CF recorded before NBQX application (blue). (**f**) Same as in (**e**) but for a different AN fiber (ANF3): CF changed from 3 to 0.9 kHz. (**g**) Monitoring of the mechanical cochlear state during the experiment in (**e** and **f**) at two relevant frequencies, using Distortion Product OtoAcoustic Emissions (DPOAEs) and compound receptor potentials (cochlear microphonic, CM). Amplitudes of DPOAEs are shown in dB SPL; CM

*Figure 4 continued on next page*

*Figure 4 continued*
amplitudes are in dB and are normalized to the level measured before application of NBQX (measurements 1–2). CM and DPOAEs are relatively stable after the NBQX injection but vanish postmortem. The measurement noise floors (dash-dotted lines) are stable over the entire experiment.

The online version of this article includes the following source data and figure supplement(s) for figure 4:

**Source data 1.** Auditory Nerve Data NBQX.

**Figure supplement 1.** Influence of applied toxins on auditory nerve (AN)-fiber responses.

and tail, consistent with different activation modes in the two response regions. Blocking of synaptic transmission would be expected to affect thresholds independent of sound frequency.

It is conceivable that the loss of sensitive tips as in *Figure 4b, e and f* does not reflect a loss of synaptic drive but rather a decline in cochlear health and sensitivity. Even if such a functional cochlear decline is present, we still expect NBQX to block synaptic transmission and eliminate or reduce sound responses. The observed remaining response is thus consistent with a contribution by intrinsic mechanical sensitivity of nerve fibers, independent of the source of the loss of the tip of frequency tuning. Moreover, as assessed by cochlear emissions and cochlear microphonics monitored throughout the experiment, the response in *Figure 4a–f* was from healthy cochlea. Acoustic distortion products elicited by tones spanning the CF range of the two fibers illustrated in *Figure 4e and f* remained high after injection of NBQX, as did the cochlear microphonic at similar frequencies (*Figure 4g*). Both signals disappeared when the experiment was terminated by anesthetic overdose. These observations support the interpretation that, even though the OC's underlying vibrations were intact, the synaptic blocker abolished the tuning curve's tip because it only depends on synaptic input. Simultaneously, the tail is more resistant because it reflects a synaptic drive combined with intrinsic mechanical sensitivity.

It is also conceivable, albeit unlikely that synaptic activation is qualitatively different at high- and low-sound levels so that a competitive blocker such as NBQX does not effectively block synaptic transmission at high-sound levels. The effects of nimodipine and isradipine L-type $Ca^{2+}$ channel and synaptic exocytosis blockers (*Huang and Moser, 2018*; *Rodriguez-Contreras and Yamoah, 2001*) were also tested. *Figure 4*, *Figure 4—figure supplement 1* shows tuning curves measured before and after the injection of nimodipine through the RW. There is a profound suppression of spontaneous activity for recording depths up to ~500 µm, accompanied by very shallow tuning curves, often consisting of only tails (*Figure 4*, *Figure 4—figure supplement 1*). Deeper in the nerve, spontaneous rates and shapes of tuning curves are less profoundly affected, likely due to limitations in the diffusion of the drug to more apical cochlear regions. The application of isradipine produced similar effects.

## Discussion

AN fibers' encoding responses are remarkably similar to other sensory neurons, with short-duration APs and inherently delayed neurotransmitter mechanisms (*Glowatzki and Fuchs, 2002*; *Griesinger et al., 2005*; *Li et al., 2014*). It has been a paradox how the AN utilizes conventional neural mechanisms to operate at 100- to 1000-fold faster time scales with acute precision and reliability than other neural systems (*Hudspeth, 1997*; *Köppl, 1997*). Puzzling auditory features such as multi-component frequency tuning and phase locking may serve as gateways to gain mechanistic insights. Motivated by the finding that AN neurons respond to minute mechanical displacement, an investigation was conducted to show whether type I SGNs are mechanically sensitive at their unmyelinated nerve endings in addition to using established pathways for neurotransmission.

It was discovered that simultaneous current injection (simulating the effect of neurotransmission) and mechanical displacement could interact to affect AN firing rates and phase-locked responses. Combined current injection and mechanical activation of the AN affect the phase angle at which spikes are fired and can generate peak-splitting and spike rate increase and suppression. In vivo recordings from chinchilla and cat demonstrate that blocking the AN activation pathway via neurotransmission does not eliminate sound-evoked AN spiking at high sound levels. Therefore, it is proposed that sound-evoked spiking in the AN, particularly at high-sound levels, is buoyed by the intrinsic mechanical sensitivity of AN fibers and that this sensitivity underlies response properties that have not been well understood (*Kiang, 1990*). The use of presynaptic neurotransmission and postsynaptic mechanical

transmission may be a cochlear feature that sculpts the extraordinary auditory response (*Griesinger et al., 2005*). Mechanoreception in AN may also serve developmental and regenerative features.

In development, mechanical interaction-mediated axon pathfinding to their target is one mechanism by which growth cones engage the extracellular matrix (*Kilinc et al., 2015*). GsMTx4, a selective blocker for mechanosensitive channels, promotes neural outgrowth (*Jacques-Fricke et al., 2006*), implicating the role of mechanics in neuronal development and neural extension following injury. Whether the current findings reflect AN developmental or functional phenomena may require future studies, but the two may not be mutually exclusive. A previous study using spider (*Argiope trifasciata*) venom showed the complete absence of AN activity even at high SPL (*Cousillas et al., 1988*), which contradicts our findings. However, unfractionated spider venom, as used in that study, contained many polyamines, polypeptides, and proteins, known to block glutamate receptors and TRP channels (e.g., GsMTx4) (*Bowman et al., 2007*; *Escoubas et al., 2000*; *Zhelay et al., 2018*). Thus, the complete absence of spiking activity may have resulted from components of the venom blocking AN mechanical sensitivity, in addition to its synaptic effects. A total absence of spiking activity has also been reported after severe acoustic trauma (*Robertson, 1982*), but only when accompanied by extensive destruction of the OC. Such destruction is bound to affect the mechanical microenvironment of AN dendrites. Particularly when combined with recent insights into damage of AN dendrites by acoustic trauma (*Liberman, 2017*), it remains unclear whether the absence of spiking as observed by *Robertson, 1982*, supports or refutes our hypothesis of a role of intrinsic mechanical sensitivity in AN fibers in the healthy cochlea.

The in vitro recordings were made using soma and dendritic substrate displacement as a proxy for AN dendritic terminal response to OCM. In particular, AN sensitivity derived from soma displacement (1.4 μm/nA slope: *Figure 2*) may underestimate the response properties at the terminals. Nonetheless, assuming ~200 nm saturating receptor potential (*Russell and Kössl, 1992*), ~145 pA would ensue, substantially altering the total synaptic potential, contributing to the synaptic output. The difference in sensitivity between apical versus basal SGNs may reflect the expected range of motions along the cochlear contour.

The findings from this report provide evidence for AN intrinsic mechanical sensitivity. If the AN's presynaptic nerve endings directly sense sound-mediated OCM, then the distinctive classical role of the AN as a primary neuron requires reevaluation. Moreover, the findings are in keeping with the emerging evidence that primary afferent neurons, such as visual retinal ganglion cells and DRG neurons, utilize melanopsin and mechanosensitive channels, respectively, to modulate sensory processing (*Coste et al., 2010*; *Hattar et al., 2002*).

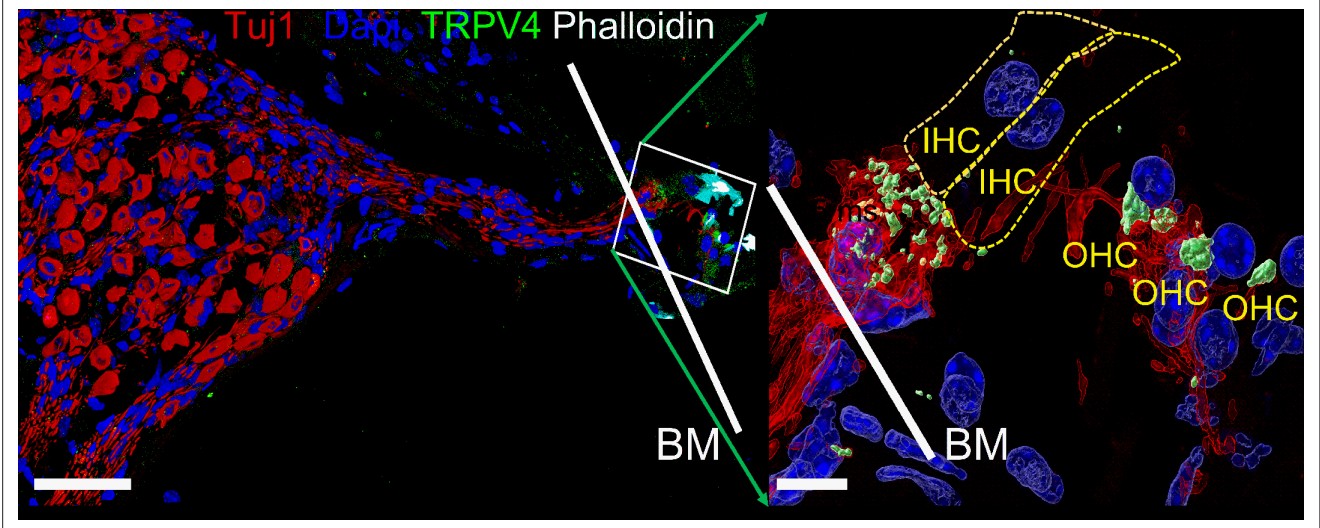

**Figure 5.** Expression of transient receptor potential vanilloid IV (TRPV4) in spiral ganglion neurons (SGNs). Positive reactivity of SGN nerve terminals to TRPV4 antibody (green). Reaction at the cell body (soma) was low. *Table 4* provides a summary of TRP channels tested and a qualitative evaluation. Scale bar low magnification (left panel = 30 μm), high magnification (right panel = 5 μm).

**Table 4.** Expression of transient response potential (TRP) channel in spiral ganglion neuron (SGN) soma and nerve terminals.
The table shows a summary of immunofluorescent detection of TRP channels at SGN cell body (soma) and nerve terminals.

| Channel | Soma | Nerve terminals |
|---------|------|-----------------|
| TRPA1 | - | - |
| TRPC3 | - | ++ |
| TRPC6 | ++ | + |
| TRPV1 | - | - |
| TRPV4 | + | +++ |

- = no reactivity.

+, ++, and +++ , = low, medium, and high reactivity, respectively.

## Mechanical sensitivity of auditory neurons in vitro

The present study's data show that SGN cell-body or dendrite displacement activates a membrane conductance with a reversal potential ~0 mV, suggesting a nonspecific cationic current. The displacement-induced current and membrane depolarization sensitivity to GsMTx4 block (*Bae et al., 2011*) further indicates that SGNs express mechanically sensitive channels. It was not the aim of this study to identify the specific candidate mechanically sensitive pathway; however, several nonspecific cationic channels such as the transient receptor potential type 3 (TRPC3) (*Phan et al., 2010*), vanilloid I (TRPV1) (*Zheng et al., 2003*), and polycystine (TRPP2) (*Takumida and Anniko, 2010*) membrane proteins have been identified in SGN. Other putative mechanically gated ion channels in SGNs can be derived from the library of single-neuron RNA-sequence analyses (*Shrestha et al., 2018*; *Sun et al., 2018*). These mechanically gated channels include Piezo-1 and Piezo-2, identified in HCs and other cochlear cell types (*Beurg and Fettiplace, 2017*; *Corns and Marcotti, 2016*; *Shrestha et al., 2018*). We detected SGN nerve terminal positive reactivity toward TRPV4 (*Figure 5*) and other TRP channels (*Table 4*). Null mutation of *Piezo*-2 in mice decreases the sensitivity of auditory brainstem responses by ~20 dB. It is unlikely that Piezo-2-mediated currents in HCs account for the auditory phenotype seen in the null-mutant mouse. The Piezo-2-mediated current in HC declines during development and may be functionally insignificant after hearing (*Beurg and Fettiplace, 2017*). The identity of the well-studied, pore-forming mechanically gated channel in HCs remains unknown (*Giese et al., 2017*; *Qiu and Müller, 2018*). If SGNs utilize a distinct mechanically gated channel, a search for the gene and protein may include emerging single-cell transcriptomic and phenotypic analyses of mouse models, awaiting future studies.

Some conditions of combined (current + displacement) stimulation (*Figure 3—figure supplement 2*) and even of displacement by itself (*Figure 1—figure supplements 1; 2 and 6*) caused suppression of spike rate of SGNs. While the data presented here cannot directly address the underlying mechanism for how two inwardly directed currents (EPSC and $I_{MA}$) yield suppressive effects, reasonable explanations can be considered. Recent reports have shown that besides voltage dependence, outward $K^+$ channels such as $K_v1.1/1.2$ and $K_v7$ are modulated by mechanical stimuli (*Hao et al., 2013*; *Perez-Flores et al., 2020*), whereby the channel's voltage sensitivity is enhanced (*Long et al., 2005*). $K_v1$ and $K_v7$ channels are members of the cadre of outward $K^+$ channels that dominates the type I afferent AN membrane (*Lv et al., 2010*; *Mo et al., 2002*; *Wang et al., 2013*). If the magnitude of outward $K^+$ currents enhanced by mechanical displacement exceeds the mechanically activated inward currents, the resulting outcome would be membrane repolarization, which would suppress AN activity, consistent with the current results.

The suppression of AN fibers' spiking is a known phenomenon in vivo and has been observed under two conditions. First, a sound can suppress the response to another sound ('two-tone suppression') (*Sachs and Kiang, 1968*). While this is partly grounded in cochlear mechanical vibration, it remains unclear whether other mechanisms contribute (*Robles and Ruggero, 2001*; *Versteegh and van der Heijden, 2013*). Second, suppression of spiking can occur even in response to single, low-frequency tones of increasing intensity. Such stimuli trigger responses with an increasing number of spikes that are phase-locked at a preferred stimulus phase, but at sound levels of 80–90 dB SPL, a set of associated changes abruptly occur: firing rate drops over a narrow range of intensities ('Nelson's notch') (*Kiang and Moxon, 1972*; *Liberman and Kiang, 1984*), and this is accompanied by a change in phase locking toward multiple preferred phases (peak-splitting). At higher intensities, the response returns to a high spike rate and monophasic phase locking, albeit at values that differ from those

at lower intensities. One functional consequence is increased coding of envelope fluctuations at high sound levels (*Joris and Yin, 1992*). Peak-splitting suggests an interaction of two pathways with different growth functions, which sum at the level of the AN (*Kiang, 1990*; *Kiang et al., 1986*). We observed phenomenologically similar events in vitro when combining sinusoidal current injection and displacement at varying relative phases. Some combinations generated peak-splitting, accompanied by decreases in firing rate (*Figure 3—figure supplement 2*). These findings suggest that the mechanical sensitivity of SGNs should be considered a possible factor in the suppressive and peak-splitting phenomena observed in vivo.

Technical limitations restricted in vitro experiments to much lower frequencies (by about a factor of 10) than those at which peak-splitting is typically studied in vivo. Note that peak-splitting becomes more prevalent and occurs over a broader range of sound levels when the sound frequency is lowered to values similar to those used here in vitro (*Oshima and Strelioff, 1983*; *Ronken, 1986*; *Ruggero and Rich, 1983*). We surmise that mechanical displacement or deformation of SGN dendrites affects their response to sound over a broad range of frequencies, but with a bias toward low frequencies. Moreover, mechanical effects are bound to occur at a particular phase relationship relative to the synaptic drive on the same dendrite, and that relationship likely varies with stimulus frequency and possibly also with the cochlear longitudinal location. The relative magnitude and phase of synaptic versus mechanical stimulation would determine the phase and probability of spiking.

## Mutually interacting elements of neurotransmission and mechanical activation at the first auditory synapse

Besides the complexities in phase locking that AN fibers show in vivo in response to simple tones, complexities in frequency tuning are observed as well: both sets of phenomena point to the existence of multiple components driving AN responses (*Kiang, 1990*; *Kiang et al., 1986*; *Kiang and Moxon, 1972*; *Liberman and Kiang, 1984*). In the cat, the species in which this was first and most extensively described, a characterization in terms of two components was proposed, with one component dominating at low sound levels and a second component at high sound levels (*Kiang, 1990*; *Liberman and Kiang, 1984*). Similar phenomena, particularly regarding phase locking, have been reported in other species but were not always restricted to high sound levels. The source of these components is controversial but has been sought at the level of cochlear mechanics or HCs, not the AN (*Cai and Geisler, 1996*; *Cheatham and Dallos, 1998*; *Cody and Mountain, 1989*; *Dallos, 1985*; *Heil and Peterson, 2019*; *Kiang, 1990*; *Liberman and Kiang, 1984*; *Nam and Guinan, 2016*; *Nam and Guinan, 2018*; *Ruggero and Rich, 1983*; *Ruggero et al., 1986*; *Russell and Kössl, 1992*).

We hypothesized that intrinsic mechanical sensitivity contributes to the high-threshold, 'tail' region of tuning curves, which is less vulnerable to a range of cochlear manipulations than the tip (*Kiang et al., 1986*; *Liberman and Kiang, 1984*). These tails may be associated with clinically relevant phenomena, such as recruitment of middle ear reflexes and abnormal growth of loudness after acoustic trauma (*Liberman and Kiang, 1984*). Delivery of synaptic blockers diminished or abolished spontaneous activity and the tip region of the tuning curve. The observation that AN responses persisted at high sound levels is in keeping with the prediction that a second mode, direct mechanical activation of AN fibers, operates and dominates high-threshold segments of the tuning curve (*Figure 4*, *Figure 4—figure supplement 1*).

Suppose neurotransmitter release-mediated EPSCs are the sole driver of AN responses as suggested (*Fuchs et al., 2003*) and suppose the mechanisms underlying the peculiarities in frequency tuning precede the synapse between IHC and AN fiber. In that case, synaptic blockers should attenuate AN activity at all intensities and should not differentially affect the presence of multiple components. This was not what we observed: while low-threshold responses were abolished, spikes could still be elicited at high intensities (*Figure 4a–f*), even in conditions where cochlear mechanical sensitivity and mass receptor potentials were normal (*Figure 4g*). It is plausible that with strong depolarization, and by extension, high sound levels, the copious release of glutamate at the synapse may render a competitive antagonist, such as NBQX, ineffective (*Sheardown et al., 1990*). Arguing against this explanation is that similar results were obtained with an L-type $Ca^{2+}$ channel blocker (*Figure 4*, *Figure 4—figure supplement 1*). NBQX and $Ca^{2+}$ channel blockers' differential effects on the two components of AN tuning curves (*Figure 4*, *Figure 4—figure supplement 1*) argue against

a single-element (neurotransmitter) mechanism (*Fuchs et al., 2003*; *Glowatzki and Fuchs, 2002*). The results are consistent with two mutually interacting mechanisms: neurotransmission and direct mechanical activation.

It may be argued that bi-modal neurotransmitter release properties perhaps explain the two components of AN tuning curves. Under certain conditions, IHC synaptic release mechanisms may consist of two distinct modes (*Grant et al., 2010*), observed at steady state and which contribute to spontaneous APs in afferent fibers. Suppose the two modes of synaptic transmission represent the two components of the AN tuning curve, again. In that case, NBQX and $Ca^{2+}$ channel blockers are expected to affect both components (*Glowatzki and Fuchs, 2002*; *Grant et al., 2010*), which is not what we observed: NBQX and $Ca^{2+}$ channel blockers suppressed the sharp-frequency tip at the CF, with a remaining response showing very coarse tuning (*Figure 4*, *Figure 4—figure supplement 1*). Thus, it is unlikely that the biphasic modes of transmitter release mechanisms, or, for that matter, any synaptic or presynaptic mechanisms, account for the differential effect of blockers on the low- and high-threshold components of the AN tuning curve. We propose, therefore, that the element of the tuning curve that remains impervious to the synaptic transmission block is driven mainly by the mechanical activation of the AN. Nonspecific blockers of mechanically gated ion channels, such as $Gd^{3+}$ (*Ranade et al., 2015*), cannot be used to suppress SGN mechanical sensitivity selectively. Moreover, diffusional constraints for large molecular-weight channel blockers (e.g., GsMTx4) also preclude in vivo selective blocking of SGN mechanical sensitivity.

## Mechanical sensitivity of auditory neurons in vivo

The in vivo approach faced experimental constraints. Unavoidably, blind recording from the nerve trunk biases against neurons lacking spontaneous activity and low-threshold responses. The most severe experimental deficiency is the diffusion of blockers through long and narrow membrane-bound spaces to reach the IHCs and dendritic endings of SGNs. The observation that apical neurons tuned to low frequencies were invariably less affected than neurons innervating more basal parts of the cochlea suggests that the drugs delivered through the RW did not always reach the HC-AN synapse at more apical locations. Diffusional limits also prevented the use of larger molecules, such as GsMTx4 (*Sadreev et al., 2019*; *Verschooten et al., 2015*).

The in vitro results suggest that synaptic and mechanical effects do not merely superimpose but interact nonlinearly. For example, if one modality (current injection or mechanical deformation) is subthreshold, it can be made suprathreshold by adding the other modality (*Figure 2*). This complicates the interpretation of the in vivo experiments – a complete absence of spiking after blocking synaptic transmission does not rule out a role for intrinsic mechanical sensitivity, and a partial effect of synaptic blockers on spiking does not prove such a role. Therefore, our in vivo experiments do not conclusively demonstrate that responses at high sound levels depend on intrinsic mechanical sensitivity, but they are consistent with that proposal.

Although the thresholds of substrate displacement needed to trigger spiking in vitro are within the range of displacements of cochlear structures measured in vivo at high sound levels (*Cooper and Rhode, 1995*; *He et al., 2018*; *Lee et al., 2016a*), it is at present unclear what the displacement amplitudes are in vivo at the unmyelinated dendritic terminals of the AN. Recent measurements with optical coherence tomography reveal a microstructure of displacements, with the largest displacements occurring at low frequencies at locations away from the basilar membrane (*Cooper et al., 2018*; *He et al., 2018*; *Lee et al., 2015*). Undoubtedly, the displacement of the substrate in our in vivo experiments is only a coarse approximation of the in vivo OCM. The relevant physical stimulus is possibly in small pressure differences, rather than displacements, that follow the acoustic waveform between the OC and the modiolar compartment and cause deformations of dendrites of AN fibers (*Karavitaki and Mountain, 2007*) and trigger intrinsic mechanical effects. Sensitivity of AN fibers to displacements or pressure differences suggests new treatment strategies for the hearing impaired. Conventional cochlear implants to remediate severe deafness can be highly successful. Still, their outcome is highly variable (*Peterson et al., 2010*), so new strategies (infrared, optogenetic, piezoelectric *Inaoka et al., 2011*; *Richter et al., 2011*; *Wrobel et al., 2018*) are being pursued to supplement or replace direct electrical stimulation of the AN. Our findings suggest that a mechanical interface can provide an additional activation mode to supplement current strategies.

## Materials and methods

All in vivo experiments were performed under a protocol approved by the University of Leuven's animal ethics committee complying with the European Communities Council Directive (86/609/EEC). At the University of Nevada, Reno (UNR), experiments were performed according to the Institutional Animal Care and Use Committee guidelines of UNR. In vitro experiments were performed using mice and chinchilla. Equal numbers of adult male and female mice and chinchillas were used (when odd numbers were reported, the females outnumbered males). In vivo experiments were performed on adult wild-type chinchilla (*Chinchilla lanigera*) of both sexes, free from a middle ear infection and weighing between 200 and 400 g.

### Cell culture

SGNs were isolated from 5- to 8-week-old C57BL/6J mice (20–30 g) (Jackson Laboratory) and 1- to 2-month-old chinchilla (200–300 g) as described previously (*Lee et al., 2016b*). The age range for mice was selected since the C57 strain shows an early hearing loss (*Spongr et al., 1997*). The apical and basal cochlea SGNs were dissociated using a combination of enzymatic and mechanical procedures. The neurons were maintained in culture for 2–5 days. Animals were euthanized, and the temporal bones were removed in a solution containing Minimum Essential Medium with HBSS (Invitrogen), 0.2 g/l kynurenic acid, 10 mM $MgCl_2$, 2% fetal bovine serum (v/v), and 6 g/l glucose. The central spiral ganglion tissue was dissected and split into three equal segments: apical, middle, and basal, across the modiolar axis, as described previously (*Glowatzki and Fuchs, 2002*). The middle turn was discarded, and the apical and basal tissues were digested separately in an enzyme mixture containing 1 mg/ml collagenase type I and 1 mg/ml DNase at 37°C for 20 min. We performed a series of gentle trituration and centrifugation in 0.45 M sucrose. The cell pellets were reconstituted in 900 μl of culture medium (Neurobasal-A, supplemented with 2% B27 (v/v), 0.5 mM L-glutamine, and 100 U/ml penicillin; Invitrogen), and filtered through a 40 μm cell strainer for cell culture and electrophysiological experiments. For adequate voltage-clamp and satisfactory electrophysiological experiments, we cultured SGNs for ~24–48 hr to allow Schwann cells' detachment from neuronal membrane surfaces.

Chinchillas were purchased from Moulton Chinchilla Ranch (Rochester, MN). Chinchilla SGNs were isolated using a protocol similar to that employed for mice. The tendency for neuronal culture from the chinchilla to profusely generate glial cells was high. We inhibited glial-cell proliferation using 20 μM cytosine arabinoside (AraC; Sigma) (*Schwieger et al., 2016*). Electrophysiological experiments were performed at room temperature (RT; 21–22°C). Reagents were obtained from Sigma Aldrich unless otherwise specified.

### Electrophysiology

Experiments were performed in standard whole-cell recording mode using an Axopatch 200B amplifier (Axon Instruments). For voltage-clamp recordings patch pipettes had resistance of 2–3 MΩ when filled with an internal solution consisting of (in mM): 70 CsCl, 55 NMGCl, 10 HEPES, 10 EGTA, 1 $CaCl_2$, 1 $MgCl_2$, 5 MgATP, and 0.5 $Na_2GTP$ (pH adjusted to 7.3 with CsOH). The extracellular solution consisted of (in mM): 130 NaCl, 3 KCl, 1 $MgCl_2$, 10 HEPES, 2.5 $CaCl_2$, 10 glucose, and 2 CsCl (pH was adjusted to 7.3 using NaOH). For current-clamp recordings, pipettes were filled with a solution consisting of (in mM) 134 KCl, 10 HEPES, 10 EGTA, 1 $CaCl_2$, 1 $MgCl_2$, and 5 MgATP and 0.5 $Na_2GTP$ (pH 7.3 with KOH). Currents were sampled at 20–50 kHz and filtered at 2–5 kHz. Voltage offsets introduced by liquid junction potentials (2.2 ± 1.2 mV [n = 45]) were not corrected. Leak currents before mechanical stimulations were subtracted offline from the current traces and were <20 pA. Recordings with leak currents greater than 20 pA were discarded. A stock solution of 10 mM GsMTx4 (CSBio; Menlo Park, CA) was prepared in water.

### Mechanical stimulation

Mechanical stimulation was achieved using a fire-polished and sylgard-coated glass pipette (tip diameter ~1 μm), positioned at ~180° to the recording electrode. The probe's movement toward the cell was driven by a piezoelectric crystal micro stage (E660 LVPZT Controller/Amplifier; Physik Instruments). The stimulating probe was typically positioned close to the cell body without visible membrane deformation. The stimulation probe had a velocity <20 μm/ms during the ramp segment of the command for forwarding motion, and the stimulus was applied for a duration, as stated in

each experiment. We assessed mechanical sensitivity using a series of mechanical steps in ~0.14 μm increments applied every 10–20 s, which allowed for the full recovery of mechanosensitive currents between steps. $I_{MA}$ were recorded at a holding potential of –70 mV. For instantaneous I-V relationship recordings, voltage steps were applied 8 s before the mechanical stimulation from holding potentials ranging from –90 to 90 mV.

SGNs were cultured on a PDMS substrate treated with poly-D-lysine (0.5 mg/ml) and laminin (10 mg/ml) to test for the mechanosensitivity of nerve endings. A single neurite can be stretched by substrate indentation on this platform without contacting the neurite (*Figure 1b* inset). The whole-cell patch-clamp recording was used to examine the electrical response to neurite stretching conducted with our direct approach of indentation of a PDMS substrate at a location adjacent to the neurite with a pipette. To study the neurons' firing response to time variation with simultaneous sine wave current and mechanical stimulation, the mechanical stimulus phase was shifted in 45° steps from 0 to 315° relative to the current. Positive voltage to the actuator corresponds to downward displacement. Technical limitations restricted in vitro experiments sinusoidal mechanical stimulation to ~100 Hz and current injection to ~1000 Hz.

## Cryosection

The temporal bones were removed and fixed in 4% paraformaldehyde in phosphate-buffered saline (PBS) for 1.5 hr at 4°C. The temporal bones were decalcified by incubation in 10% EDTA at 4°C for 3–5 days. The EDTA solution was changed daily. The bones were then embedded in the OCT compound for cryostat sectioning. The sections of 10 μm thickness were washed in PBS, and nonspecific binding was blocked with 1% bovine serum albumin (BSA) and 10% goat serum in PBS plus 0.1% Triton X-100 (PBST) for 1 hr. The primary antibodies, chicken anti-Tuj1 (Abcam), mouse anti-myelin basic protein (Abcam), rabbit anti-Myo7a (Proteus Biosciences, Inc), mouse anti-Tuj1 (Abcam), rabbit anti-TRPA1 (Abcam), TRPC3 (Novus Biologicals), TRPC6 (Abcam), TRPV1 (Novus Biologicals), TRPV4 (Abcam), were incubated overnight at 4°C. After incubating the primary antibodies, the slides were washed three times with PBST and incubated with secondary antibodies for 1.5 hr at RT in the dark. We used Alexa Fluor 647-conjugated goat anti-mouse and Cy3-conjugated goat anti-chicken, Alexa Fluor 488-conjugated goat anti-rabbit, Alexa Fluor 568-conjugated goat anti-mouse (Jackson ImmunoResearch Labs) in a dilution of 1:500. Other markers used were phalloidin-Fluor 647 (Abcam) for F-actin and DAPI (Sigma) for nuclear stain. The slides were then examined under a confocal microscope (LSM 510, Zeiss).

## Data analysis

Data analyses were performed offline using pClamp8 (Axon Instruments) and Origin software (Microcal Software, Northampton, MA). Statistical analysis was performed using a paired or unpaired t-test, with significance at $p < 0.05$. The peak $I_{MA}$ for each step displacement was expressed in channel-open probability ($p_o$) and plotted against displacement (X). The relationship was fitted with a one-state Boltzmann equation $p_o = 1/[1 + e^{z(X - X_{0.5})/(kT)}]$ to obtain channel gating force, $z$, and the displacement at 50% open probability ($X_{0.5}$), and T is temperature. For a two-state Boltzmann function $p_o = 1/([1 + e^{za(X - X_{0.5a})/(kT)}] + 1/[1 + e^{zb(X - X_{0.5b})/(kT)}])$. The dose-response relationships were described with a logistic function; $(I_i − I_f)/(1 + [C]/[C_{0.5}])p + I_f$. $I_i$ and $I_f$ are the initial and final magnitudes, [C] is the drug's concentration with [$IC_{0.5}$] its half-blocking concentration, and p is the Hill coefficient. The decay phases of the $I_{MA}$ were fitted by a bi-exponential decay function of the form: $y(t) = A_1 * \exp(−t/\tau_1) + A_2 * \exp(−t/\tau_2) + A_{ss}$, where t is time, $\tau$ is the time constant of decay of $I_{MA}$, $A_1$ and $A_2$ are the amplitudes of the decaying current components, and $A_{ss}$ is the amplitude of the steady-state, non-inactivating component of the total $I_{MA}$. The strength of phase locking was quantified as VS, which is the ratio of the period histogram's fundamental frequency component to the average firing rate (*Goldberg and Brown, 1969*). Also known as the synchronization index, the VS varies from VS = 0 for a flat histogram with no phase locking to VS = 1 for a histogram indicating perfect phase locking. Statistical significance of the VS was assessed by Rayleigh statistics (*Mardia, 1972*), using a criterion of $p < 0.05$; values of VS failing this criterion were discarded. Data are presented as mean ± SD (standard deviation).

## Surgical preparation for in vivo experiments

In chinchillas, anesthesia was initiated by intramuscular (im) injection of a ketamine-xylazine mixture and was maintained with ketamine and diazepam, titrated according to vital signs and reflexes. In cats, induction was with a 1:3 mixture of ketamine and acepromazine and maintenance with intravenous infusion of pentobarbital. The animal was placed on a feedback-controlled heating pad in a double-walled sound-proof room. A tracheotomy was performed, and the respiration rate and end-tidal $CO_2$ were continuously monitored. The acoustic system was placed in the external auditory meatus and calibrated with a probe microphone. The AN was accessed via a traditional posterior fossa approach in which a small portion of the lateral cerebellum was aspirated. The tympanic bulla on the recording side was opened to visualize the cochlear RW for cochlear potential measurements and blockers' administration. Details of our procedures are available elsewhere (*Bremen and Joris, 2013*; *Louage et al., 2006*).

## Acoustic stimulation and recording

Acoustic stimulation and acquisition of signals utilized custom software to control digital hardware (RX6, Tucker-Davis Technologies, Alachua, FL). Stimuli were compensated for the transfer function of the acoustic system. Acoustic stimuli were delivered through dynamic phones. For CM and DPOAE recordings, analog signals were recorded (RX6) and analyzed offline (MATLAB, MathWorks, Natick, MA). CM recordings were obtained with a silver ball electrode near the RW. The reference and ground wire electrodes from the differential amplifier were placed in the skin next to the ear canal and in the neck's nape, respectively. The signal was amplified with a differential amplifier (RS560, Stanford Research Systems), recorded (RX8, Tucker-Davis Technologies, Alachua, FL), stored on a computer, and processed with custom software in MATLAB. The CM was obtained for different stimulus frequencies (2, 3, 4, 6, 8, 10, 12 kHz) and was spectrally calculated from the difference of the evoked responses (divided by 2) to alternating pure tones. DPOAE responses were recorded and stored using the same microphone and acquisition system for the acoustic calibration. The primaries (F1 and F2, duration 700 ms, repetition interval 800 ms) were generated with separate acoustic actuators to minimize actuator distortion products. The frequency ratio (F2/F1) and amplitude levels (L1, L2) of the primaries were fixed to a ratio of 1.21 and 65 (L1) and 55 (L2) dB SPL, respectively. DPOAEs were quantified as the sound pressure of the returning 2F1-F2 cubic difference distortion product.

Single fiber recordings were obtained with micropipettes filled with 2 M KCl (impedance ~40–80 MΩ), mounted in a hydraulic micro-drive, and placed above the nerve trunk under visual control. Zero depth was marked at initial contact. If necessary, warm agar (2%) was poured on the AN to reduce pulsations. The neural signal was recorded and displayed using routine methods. An intracellular amplifier (Dagan BVC-700A) allowed monitoring of DC-shifts signaling axonal penetration and enabled the use of current pulses to improve the recording.

Further amplification and filtering (~100Hz to 3 kHz) preceded the conversion of spikes to standard pulses with a custom peak-detector (1 μs resolution) and internal RX6 timer. A search stimulus consisting of 60 dB SPL, 200 ms tone pips stepped in frequency was used while the microelectrode was advanced in 1 μm steps until large monopolar APs were obtained. After the blocker administration, the search stimulus's sound level was gradually increased, and the duration of the tone pips reduced to minimize cochlear trauma. For each isolated AN-fiber, the spontaneous rate (SR: estimated over a 15 s interval) was measured, and a threshold tuning curve was obtained with a tracking program using short (30 ms) tone bursts (repetition intervals 100 ms; rise-fall time 2.5 ms). We extracted the CF as the frequency of the lowest threshold from the threshold tuning curve.

## Acknowledgements

We thank members of our laboratories for their comments on this manuscript. We thank Dr Frederick Sachs for his constructive comments and suggestions on the first draft of the script. Grants to ENY supported this work from the National Institutes of Health (DC016099, DC015252, DC015135, AG060504, AG051443). PXJ was supported by grants from KU Leuven (BOF, OT-14-118) and Fonds Wetenschappelijk Onderzoek (Research Foundation Flanders) G0B2917N.

# Additional information

## Funding

| Funder | Grant reference number | Author |
|---|---|---|
| National Institutes of Health | DC016099 | Ebenezer N Yamoah |
| National Institutes of Health | DC015252 | Ebenezer N Yamoah |
| National Institutes of Health | DC015135 | Ebenezer N Yamoah |
| National Institutes of Health | AG060504 | Ebenezer N Yamoah |
| National Institutes of Health | AG051443 | Ebenezer N Yamoah |
| KU Leuven | OT-14-118 | Philip X Joris |
| Fonds Wetenschappelijk Onderzoek | G0B2917N | Philip X Joris |
| Fonds Wetenschappelijk Onderzoek | G085421N | Philip X Joris |

The funders had no role in study design, data collection and interpretation, or the decision to submit the work for publication.

## Author contributions

Maria C Perez-Flores, Data curation, Formal analysis, Investigation, Methodology, Writing – review and editing; Eric Verschooten, Data curation, Formal analysis, Investigation, Methodology, Validation, Writing – review and editing; Jeong Han Lee, Data curation, Formal analysis, Investigation, Methodology; Hyo Jeong Kim, Formal analysis, Investigation, Methodology, Software; Philip X Joris, Conceptualization, Formal analysis, Funding acquisition, Methodology, Project administration, Resources, Supervision, Visualization, Writing - original draft, Writing – review and editing; Ebenezer N Yamoah, Conceptualization, Data curation, Funding acquisition, Investigation, Methodology, Project administration, Resources, Supervision, Validation, Writing - original draft, Writing – review and editing

## Author ORCIDs

Maria C Perez-Flores (iD) http://orcid.org/0000-0002-2166-5648
Eric Verschooten (iD) http://orcid.org/0000-0001-8010-363X
Philip X Joris (iD) http://orcid.org/0000-0002-9759-5375
Ebenezer N Yamoah (iD) http://orcid.org/0000-0002-9797-085X

## Ethics

This study was performed in strict accordance with the recommendations in the Guide for the Care and Use of Laboratory Animals of the National Institutes of Health. All of the animals for in vitro experiments were handled according to approved institutional animal care and use committee (IACUC) protocols (#08-133) of the University of Arizona. The protocol was approved by the Committee on the Ethics of Animal Experiments of the University of Minnesota (Permit Number: 27-2956). All surgery was performed under sodium pentobarbital anesthesia, and every effort was made to minimize suffering (Yamoah, UNR protocol). The animal procedures for in vivo experiments were in accordance with the European Communities Council Directive (86/609/EEC) and approved by the animal ethics committee of the University of Leuven.

## Decision letter and Author response

Decision letter https://doi.org/10.7554/eLife.74948.sa1
Author response https://doi.org/10.7554/eLife.74948.sa2

## Additional files

### Supplementary files
• Transparent reporting form

### Data availability
All data generated or analysed during this study are included in the manuscript and supporting file; Source Data files have been provided for Figures 1-4.

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
