## [Editor Report]

The transduction of mechanical sound signals into electrical activities has so far been attributed to the inner hair cells of the inner ear. Here the authors show that the spiral ganglion neurons that innervate the inner hair cells are mechanosensitive as well. In particular, the authors find that spiral ganglion cells maintained in cell culture can fire action potentials in response to mechanical stimulation. Moreover, the authors provide evidence that the mechanotransduction in the spiral ganglion cells may contribute to sound detection in vivo.

---

## [Decision Letter]

**Decision letter after peer review:**

[Editors’ note: the authors resubmitted a revised version of the paper for consideration. What follows is the authors’ response to the first round of review.]

Thank you for submitting the paper "Intrinsic mechanical sensitivity of auditory neurons as a contributor to sound-driven neural activity" for consideration by *eLife*. Your article has been reviewed by 3 peer reviewers, including Tobias Reichenbach as the Reviewing Editor and Reviewer #1, and the evaluation has been overseen by a Senior Editor.

We are sorry to say that, after consultation with the reviewers, we have decided that this work will not be considered further for publication by *eLife*.

The reviewers raised a range of substantial criticisms, especially questioning the relevance of the SGN's mechanosensitivy in vivo, which appears highly unlikely, not least due to previous findings by Cousillas, Cole, and Johnstone (Hear. Res. 1988).

*Reviewer #1:*

The paper from Maria Perez Flores deals with a general phenomenon of mechanical sensitivity of neurons. In the case of the neurons in the cochlea (SGNs), this mechanical sensitivity comes together with mechanical stimuli in the vicinity. The investigation of whether the mechanics in the inner ear could directly stimulate the SGN is a very exciting question. The data are clearly and comprehensibly presented in individual datasets and support the finding of a mechanical sensitivity. The assignment of the data sets to the different animal model systems needs to be made more clearly and ideas for the use and application of the results incorporated more explicitly.

I started to read the paper a few times because the topic is interesting and could be of general interest. However, the mixture of animal models and data from literature in the results part, which are not clearly kept apart and the cluttered figures have impeded my flow of reading and understanding. Therefore, before publication, the text flow and figures in particular must be improved to increase the readability.

Please go over the text and figures again and always indicate which animal group was measured and what numbers led to the data.

In the results part, an unusual mixture of new data and information from former experiments are shown (e.g. page 12 second paragraph). If really necessary to include, the authors should clearly mention when they start to present their own data.

SGNs receive input also from OHCs. This should be mentioned and discussed in the manuscript. Further, tuning curves that lost their tip can be also related to OHC-function (amplification) lost (Kiang et al., 1986, Hear Res.). How does this fit with the data? There are also DPOAE amplitude reductions of 10 dB, but detailed data are not provided.

*Reviewer #2:*

The authors present an intriguing and novel set of findings to support their view that "mechanical sensitivity interacts with synaptic responses to shape responses in the AN, including frequency tuning and temporal phase-locking. The combination of neurotransmission and mechanical sensation to control spike patterns gives the AN a secondary receptor role, an emerging theme in primary neuronal functions."

The authors present the exciting and novel finding of mechanical sensitivity in cultured adult SGNs. This alone is a very interesting finding based on well conducted measurements. I have no criticism of these but on the interpretation of the results. The outcome of these in vitro measurements and the authors' interpretation of them in the context the in vivo measurements would have been strongly supported by immunohistochemistry of the distribution of mechanoreceptors on the SGNs. A major issue concerns their presence both in vitro and in vivo. Are they present both in in vitro and in vivo? If mechanoreception is expressed in vivo are they the same as those expressed in vitro? My reason for raising this question is because it is proposed that developing and regenerating neurons use mechanoreception for pathfinding (e.g. PMID: 26283918; PMID: 16723522). Could this be what the authors are investigating in their in vitro measurements? Is the mechanoreception the authors measure in the cultured SGN still expressed in the adult cochlea? Mechanoreception can be suppressed, or expressed, depending on the matrices to which developing neurons are exposed. This question appears not to have been intensively studied in the mammalian cochlea and the authors could be able to extend their research approach to understanding how SGNs seek and form synapses with their IHC and OHC targets and at the appropriate locations on these targets. My reason for suggesting that the mechanoreception the authors measure in vitro is that it may not actually be present in vivo, according to earlier reports, which the authors may wish to consider and dispute.

The outcome of your measurements from fibers in the auditory nerve trunk contrasts with those of earlier measurements. Notably those of Cousillas, Cole, Johnstone (1988, PMID 2905359). Johnstone et al. used the technique developed by Manley and Robertson (1974) to make recordings from individual spiral ganglion cells in the spiral lamina immediately adjacent to the inner hair cells. The advantage of this technique is that Cousillas et al. knew exactly which group of IHCs formed synapses with the SGNs and they could also separate the SGNs into high and low spontaneous rate fibers. They found that a spider toxin (Argiope trifasciata a highly specific blocker of the ionic channels associated with invertebrate glutamatergic receptors) which they perfused through the scala tympani of the basal turn directly on the SGs from which responses were being measured, blocked spontaneous activity and all responses to tones at the CF of the fibers up to 130 dB SPL. Blockage of spontaneous activity depended on the spontaneous rate of the fiber. Blockade was most effective on high spontaneous rate fibers and low spontaneous rae fibers could be more effectively blocked if they were acoustically stimulated. Responses to acoustic stimulation (even at 129 dB SPL) could be completely suppressed. Johnstone et al. detected no residual mechanical sensitivity in the SGs and presented arguments, supporting earlier studies that l-glutamate was the likely transmitter released by the inner hair cell afferent synapses. There is another paper by Robertson (Hearing Res.7, 55-74, 1982) who correlated the histology of acoustic trauma with SGN responses in vivo. He made several interesting findings but in essence, if the IHCs had been damaged, the SGNs did not show mechanical sensitivity, even at levels around 130 dB SPL. My view is that the outcomes of these two papers (there are others including by Bobbin, which I have not fully read) should be addressed in the light of your reported findings, because they appear to be at odds with your findings and conclusions.

I think I have covered the main issues in my comments above. My concern is not with the quality of the measurements and their outcomes, but in their interpretation in the light of earlier investigations, which appeared not to have been noted by the authors. The in vitro study has application and significance beyond the aims of the current study. In future in vivo measurements, it might be an advantage to adopt the Manley Robertson technique, which avoids some of the uncertainties encountered by the auditory nerve measurement technique used in the paper under review.

*Reviewer #3:*

The transduction of mechanical sound signals into electrical activities has so far been attributed to the inner hair cells of the inner ear. In this manuscript the authors show that the spiral ganglion neurons that innervate the inner hair cells are mechanosensitive as well. In particular, the authors find that spiral ganglion cells maintained in cell culture can fire action potentials in response to mechanical stimulation. Moreover, the authors provide evidence that the mechanotransduction in the spiral ganglion cells may contribute to sound detection in vivo. These results are overall well described and are supported by a variety of information and data.

Some questions and concerns that I have are:

1. The authors show that the channel blocker GsMTx4 reduces the evoked current by roughly 50%. But this means that there are still further mechanotransduction channels that are not blocked by GsMTx4. Have the authors tried out other channel blockers to reduce the evoked current further?

2. The authors state that panels f and g in Figure 3 show the effects of peak splitting. However, I can't infer this information from the panels, since the spikes are so close together. I recommend the authors show the spikes in a much shorter time window, together with the stimulus waveforms, so one can compare the spike timings to the phases of the stimulations. I am also not sure how to read panels h and m, please describe the presented information in more detail.

3. As I understand it, the spiking of the spiral ganglion neurons is independent of the direction of the flow respectively of the direction of the mechanical stimulation. As a consequence, when investigating the dependency of the firing rate on the phase between the current stimulation and the mechanical stimulus, a phase shift of 180 degrees should give the same result as no phase shift. In other words, the dependence on the phase shift should be periodic with a period of 180 degrees. Is that the case?

4. The biggest uncertainty to me is the relevance of the described mechanotransduction in vivo. The authors present results from in-vivo experiments in which they block the synaptic transmission between the inner hair cell and the spiral ganglion neuron. They show that the low-frequency part of the tuning curves remains pretty much unchanged, demonstrating that this portion of the tuning curve results from a second mechanontransduction process. But I have doubts that this second mechanotransduction process is the one that the authors have characterized before, due to the frequency response. After application of the synaptic blocker, the tuning curves are unchanged at frequencies up to 1kHz, although the authors have shown that the mechanotransduction in isolated spiral ganglion neurons was attenuated above 10 Hz and absent at 1 kHz. How can these findings be reconciled with an in vivo function up to 1 kHz?

5. It appears to me that a more direct test of the in vivo significance of the mechanotransduction in spiral ganglion neurons would be the application of the channel blocker GsMTx4 in vivo. The authors have shown that this blocker reduces the mechanotransduction in spiral ganglion neurons by approximately 50%. If the application of this blocker in vivo produced an elevation of the tuning curves, this would provide further evidence for the involvement of this pathway in in-vivo sound detection.

6. The last sentence regarding new treatments for hearing-impaired persons appears out of the blue. The authors should either eliminate this sentence, or explain further how such new treatments could work.

[Editors’ note: what follows is the authors’ response to the second round of review.]

Thank you for resubmitting your work entitled "Intrinsic mechanical sensitivity of auditory neurons as a contributor to sound-driven neural activity" for further consideration by *eLife*. Your revised article has been evaluated by Barbara Shinn-Cunningham (Senior Editor) and a Reviewing Editor.

Essential revisions:

– 93-100 AN responses, which generate unimodal cycle histograms i.e., response as a function of stimulus phase (Johnson 1980, Rose et al. 1967). In contrast, for some intermediate intensities, AN fibers fire APs at two or more stimulus phases, a phenomenon referred to as "peak-splitting" (Johnson 1980, Kiang and Moxon 1972). Moreover, a typical AN frequency tuning curve consists of two components (Liberman and Kiang 1984) – a sharply tuned tip near the characteristic frequency (CF) and a low-frequency tail, which are differentially sensitive to cochlear trauma. Both of these observations suggest that more than one process may drive AN responses.

Comment: These topics are covered by recent papers (e.g Hear Res. 2016; 341: 66-78; Hear Res. 2018 Feb;358:1-9. doi: 10.1016) that involve presynaptic mechanism. Perhaps these papers should be quoted here and commented on in the discussion.

– Lines 108-111

The geometry of the course of the unmyelinated terminal segment of SGN dendrites towards the Organ of Corti (OC) suggests that this segment undergoes some degree of mechanical deformation in response to sound.

Comment: This is an important point. Please quote supporting reference.

– Lines 137-138. "subject to OC movement (OCM) (Chen et al. 2011, Jawadi et al. 2016, Karavitaki and Mountain 2007),

Comment: I have read these papers and could not see where any of the authors states explicitly that the nerve terminals are subject to OC movement. Please correct me, if I am wrong, but it appears that you have deduced this without proof. And should make clear that this is your deduction and not a finding reported in the papers. I can see mention of previous measurements of radial motion of MOC fibres during OHC contractions but the resolution of the method, stated by the authors was too weak to resolve the very small movements they observed near the edge of the reticular lamina.

– Lines 189-191 The mechanical responses of VNs were comparable to those reported for dorsal root ganglion (DRG) neurons (Finno et al. 2019, Viatchenko-Karpinski and Gu 2016).

Comment: I am not sure. Your figure 1A shows events that appear not to be all-or-nothing spikes

– Lines 227-229 Since the exact relationship in amplitude and phase between synaptic and mechanical events in the cochlea is unknown.

Comment: The phase relationships are known between SPL, middle ear, BM, RL, Hair cell and neural responses, for example, and the IHC synaptic delay has been measured in vivo to be around 1ms. If the nerve fibres were excited directly by sound stimulation, they might be expected to respond in phase with the IHC responses and not as observed for moderate to high level (Palmer and Russell, 1986 and see Rutherford et al., 2021), after a ~1ms delay.

– Line 295-377: Auditory nerve responses in the absence of synaptic transmission

Comment: I still find this section unconvincing evidence in support of AN responding to mechanical stimulation in the absence of synaptic transmission. There remains the possibility, raised previously, that the applied agents have not fully blocked afferent transmission because they have not reached the location of the synapse. This situation is exacerbated by another recent report (J Neurophysiol. 2019 Mar 1;121(3):1018-1033). In this paper the authors show that: "as sound level is increased, the cochlear origins of CAPs from tone bursts of all frequencies become very wide and their centers shift toward the most sensitive cochlear region". Although you measure single units, this finding makes the interpretation very difficult. It is more reason to make measurements from the spiral ganglia directly.

– Lines 305-307 Cochlear displacements of this magnitude have been measured in vivo, particularly towards the cochlear apex and at high sound pressure levels (SPLs), where frequency tuning is poor (Cooper and Rhode 1995, Lee et al. 2015, Lee et al. 2016a).

Comment: One should remember that a loud tone at the apex causes a much larger mechanical displacements than at the base of the cochlea and that loud tones in cochleae that are sensitive to ultrasound move far less for a given SPL than in the same spatial region of a low frequency cochlea eg gerbil. You are correct in saying that it is important to understand the nature of any mechanical stimulus that would excite ANs in vivo.

– Lines: 483-489 Peak-splitting suggests an interaction of two pathways with different growth functions, which sum at the level of the AN (Kiang 1990, Kiang et al. 1986). We observed phenomenologically similar events in vitro when combining sinusoidal current injection and displacement at varying relative phases. Some combinations generated peak-splitting, accompanied by decreases in firing rate (Figure 3, S10). These findings suggest that the mechanical sensitivity of SGNs should be considered a possible factor in the suppressive and peak-splitting phenomena observed in vivo.

Comment: In addition to Kiangs hypothesis, peak splitting has recently been attributed to presynaptic mechanisms (Nam and Guinan, Hear Res. 2016 November ; 341: 66-78.). There is support for this idea from IHC intracellular measurements, but not OHCs (Russell and Kössl, 1992, Figure 4), although these findings were obtained from measurements whose objective was not to examine the basis of Nelson's notch.

– Lines 509-514: Similar phenomena, particularly regarding phase-locking, have been reported in other species but were not always restricted to high sound levels. The source of these components is controversial but has been sought at the level of cochlear mechanics or hair cells, not the AN (Cai and Geisler 1996, Cody and Mountain 1989, Dallos 1985, Heil and Peterson 2019, Kiang 1990, Liberman and Kiang 1984, Ruggero and Rich 1983, Ruggero et 514 al 1986).

Comment: See above and perhaps include the Nam and Guinan, 2016 paper.

– Lines 516-523: We hypothesized that intrinsic mechanical sensitivity contributes to the high-threshold, "tail" region of tuning curves, which is less vulnerable to a range of cochlear manipulations than the tip (Kiang et al. 1986, Liberman and Kiang 1984). These tails may be associated with clinically relevant phenomena, such as recruitment of middle ear reflexes and abnormal growth of loudness after acoustic trauma (Liberman and Kiang 1984). Delivery of synaptic blockers diminished or abolished spontaneous activity and the tip region of the tuning curve. The observation that AN responses persisted at high sound levels is in keeping with the prediction that a second mode, direct mechanical activation of AN fibers, operates and dominates high threshold segments of the tuning curve (Figure 4, S12).

Comment: See Nam and Guinan Hear Res. 2018 February; 358: 1-9. Who suggest that changes due to low frequency tail are due to the shearing reticular membrane motion

– Lines 528-538:

Comment: Please see comment above to line 295-377

– Figure 3: the phase locking shown in panels h and m does not appear terribly strong. In particular, the distributions do not differ dramatically from a uniform one, although the differences are probably significant. To be on the safe side, I would therefore like to see a statistical test that the distributions are indeed significantly non-uniform.

– Figure 3: The authors state that "the close alignment of the two lines shows that a high firing rate is accompanied by strong phase-locking, and vice versa that peak-splitting is associated with low firing rates." I don't understand how I am suppose to see that. First, as pointed out above, phase locking looks rather weak to me. Second, where are the two cases (high and low firing rates) represented?

– Figure 3: Connected to the above, since panels h and m summarize data with and without peak splitting, shouldn't one see an effect of peak splitting in these histograms? In other words, how can peak splitting be inferred from the histograms?

– Lines 82-87: Phase locking to sound stimuli is a feature of the AN essential for sound detection, localization, and arguably for pitch perception and speech intelligibility (Peterson and Heil 2020, Yin et al. 2019). How these response features remain sustained, despite the limits of presynaptic mechanisms of transmitter release to ATP-generation, synaptic fatigue, and vesicle replenishment (MacLeod and 86 Horiuchi 2011, Stevens and Wesseling 1999, Yamamoto and Kurokawa 1970), is not fully understood.

Comment: Perhaps include the excellent review by Rutherford et al., which covers this topic (J Physiol 599.10 (2021) pp 2527-2557).

---

## [Author Response]

[Editors’ note: the authors resubmitted a revised version of the paper for consideration. What follows is the authors’ response to the first round of review.]

Reviewer #1:The paper from Maria Perez Flores deals with a general phenomenon of mechanical sensitivity of neurons. In the case of the neurons in the cochlea (SGNs), this mechanical sensitivity comes together with mechanical stimuli in the vicinity. The investigation of whether the mechanics in the inner ear could directly stimulate the SGN is a very exciting question. The data are clearly and comprehensibly presented in individual datasets and support the finding of a mechanical sensitivity. The assignment of the data sets to the different animal model systems needs to be made more clearly and ideas for the use and application of the results incorporated more explicitly.I started to read the paper a few times because the topic is interesting and could be of general interest. However, the mixture of animal models and data from literature in the results part, which are not clearly kept apart and the cluttered figures have impeded my flow of reading and understanding. Therefore, before publication, the text flow and figures in particular must be improved to increase the readability.Please go over the text and figures again and always indicate which animal group was measured and what numbers led to the data.

We have revisited the text and figures and have made substantial changes, including splitting the multiple panel figures and separating them based on animal models. The transitions between the literature and the current data have been made clear. For example, data from chinchilla in original Figure 1 is shown in S4. We have followed the line-by-line suggestions made by the reviewer to improve and increase readability. In all cases, the animal model numbers used are listed in the legend (see line 1219, page 48; line 1241, page 55).

In the results part, an unusual mixture of new data and information from former experiments are shown (e.g. page 12 second paragraph). If really necessary to include, the authors should clearly mention when they start to present their own data.

We have removed the previous data reported and only present new data that distinguish the current. For example, references to SGN classifications using current injection are now cited, and only SGN responses to mechanical stimulus are now shown (see S6; page 1253 page 57).

SGNs receive input also from OHCs. This should be mentioned and discussed in the manuscript. Further, tuning curves that lost their tip can be also related to OHC-function (amplification) lost (Kiang et al., 1986, Hear Res.). How does this fit with the data? There are also DPOAE amplitude reductions of 10 dB, but detailed data are not provided.

Type II nerve fibers indeed receive input from OHCs but are too small to record from with our micropipettes. Recordings from the nerve trunk, even with high impedance micropipettes, exclusively sample type I neurons (Liberman, Science, 1982) and it is therefore exceedingly unlikely that any of the fibers we report on would from a type II fiber, let alone the dozens of fibers illustrated in Figure 4. It is indeed the case that tuning curves can lose their tip because of loss of OHC-function: this was clearly acknowledged in the original manuscript (p. 14, bottom paragraph) and is the reason for examining cochlear emissions and cochlear microphonics. The mild reduction of emissions, combined with the stability of the cochlear microphonics, argues against loss of OHC function as the mechanism behind the loss of the tip of the tuning curve. This is extensively discussed in response to the last set of questions from this reviewer (regarding Figure 4). Also, as mentioned in the original manuscript, a critical point is that even if OHC loss is present, it is to be expected that NBQX should completely abolish synaptic transmission, which is not what we observed (see Figure 4 line 1224, page 51 and lines 513-543).

Reviewer #2:The authors present an intriguing and novel set of findings to support their view that "mechanical sensitivity interacts with synaptic responses to shape responses in the AN, including frequency tuning and temporal phase-locking. The combination of neurotransmission and mechanical sensation to control spike patterns gives the AN a secondary receptor role, an emerging theme in primary neuronal functions."The authors present the exciting and novel finding of mechanical sensitivity in cultured adult SGNs. This alone is a very interesting finding based on well conducted measurements. I have no criticism of these but on the interpretation of the results. The outcome of these in vitro measurements and the authors' interpretation of them in the context the in vivo measurements would have been strongly supported by immunohistochemistry of the distribution of mechanoreceptors on the SGNs. A major issue concerns their presence both in vitro and in vivo. Are they present both in in vitro and in vivo? If mechanoreception is expressed in vivo are they the same as those expressed in vitro? My reason for raising this question is because it is proposed that developing and regenerating neurons use mechanoreception for pathfinding (e.g. PMID: 26283918; PMID: 16723522). Could this be what the authors are investigating in their in vitro measurements? Is the mechanoreception the authors measure in the cultured SGN still expressed in the adult cochlea? Mechanoreception can be suppressed, or expressed, depending on the matrices to which developing neurons are exposed. This question appears not to have been intensively studied in the mammalian cochlea and the authors could be able to extend their research approach to understanding how SGNs seek and form synapses with their IHC and OHC targets and at the appropriate locations on these targets. My reason for suggesting that the mechanoreception the authors measure in vitro is that it may not actually be present in vivo, according to earlier reports, which the authors may wish to consider and dispute.

We have demonstrated that several TRP channels are expressed at the SGN nerve terminals using cochlear tissue (Supplement Figure 13 and Table 2). The expression is seen in freshly isolated cochlea tissue. We have also discussed the potential role of SGN's mechanical sensitivity in axon pathfinding in development. The functional and development roles need not be mutually exclusive. Regarding the developmental roles of SGN mechanical sensitivity, we suggest it is beyond the scope of the current studies. We are interested in the subject and will proceed with those studies. Please, kindly stay tuned. The references mentioned have been cited.

The outcome of your measurements from fibers in the auditory nerve trunk contrasts with those of earlier measurements. Notably those of Cousillas, Cole, Johnstone (1988, PMID 2905359). Johnstone et al. used the technique developed by Manley and Robertson (1974) to make recordings from individual spiral ganglion cells in the spiral lamina immediately adjacent to the inner hair cells. The advantage of this technique is that Cousillas et al. knew exactly which group of IHCs formed synapses with the SGNs and they could also separate the SGNs into high and low spontaneous rate fibers. They found that a spider toxin (Argiope trifasciata a highly specific blocker of the ionic channels associated with invertebrate glutamatergic receptors) which they perfused through the scala tympani of the basal turn directly on the SGs from which responses were being measured, blocked spontaneous activity and all responses to tones at the CF of the fibers up to 130 dB SPL. Blockage of spontaneous activity depended on the spontaneous rate of the fiber. Blockade was most effective on high spontaneous rate fibers and low spontaneous rae fibers could be more effectively blocked if they were acoustically stimulated. Responses to acoustic stimulation (even at 129 dB SPL) could be completely suppressed. Johnstone et al. detected no residual mechanical sensitivity in the SGs and presented arguments, supporting earlier studies that l-glutamate was the likely transmitter released by the inner hair cell afferent synapses. There is another paper by Robertson (Hearing Res.7, 55-74, 1982) who correlated the histology of acoustic trauma with SGN responses in vivo. He made several interesting findings but in essence, if the IHCs had been damaged, the SGNs did not show mechanical sensitivity, even at levels around 130 dB SPL. My view is that the outcomes of these two papers (there are others including by Bobbin, which I have not fully read) should be addressed in the light of your reported findings, because they appear to be at odds with your findings and conclusions.

We have considered the findings of Cousillas et al., and have provided our interpretation of their conclusions. While valuable at the time, the use of whole-spider venoms (unfractionated/heat-inactivated) cannot be considered specific. Components of spider venom ranged from polyamines, peptides, and polypeptides, to large-molecular weight proteins. In particular, polyamines are known blockers of glutamatergic receptors as well as TRP channels. Thus, it is conceivable that the crudespider venoms blocked both SGN AMPA receptors' mechanosensitive channels. Also, Cousillas et al. did not include any monitoring of cochlear health (e.g., CAP recordings) – which is rather surprising, because that was a standard procedure in the Perth laboratory. Particularly given that a second pipette was placed near the basilar membrane, there is a real possibility that the cochleas were mechanically compromised.

It is even more difficult to draw firm conclusions from Robertson (1982), although this is an elegant and well-executed study. Indeed, for one animal in which a very severe lesion was observed, a total lack of response was found (“type 1” responses). However, the organ of Corti was missing here altogether and was replaced by invading epithelial cells! Were there still free unmyelinated endings? Was there actually any movement possible near the habenula perforata? Also, it is not so straightforward to be sure one is recording from a neuron when it cannot be activated: one has to rely on DC-shifts, injury discharges, or response to electrical stimulation, or (most conclusive) intra-axonal labeling (these are all technical aspects we are familiar with so that we know the challenges). Additionally, recent reports have shown that acoustic overstimulation as used in Robertson 1982 causes damage not only to hair cells but also to the AN dendrites (Kujawa and Liberman, 2009). Thus, again, both haircell and AN damage are anticipated, although at the time, AN injury was not contemplated. Given the long interval between overstimulation and recording (at least 3 weeks and sometimes much longer), it is unsure what the structural integrity of the unmyelinated part of the auditory nerve fibers was. In summary, not finding responses after strong damage is not straightforward to interpret, particularly in light of our finding of a nonlinear interaction between direct mechanical and sound stimulation.

We now briefly refer to and discuss these two papers (page 16 line 400).

I think I have covered the main issues in my comments above. My concern is not with the quality of the measurements and their outcomes, but in their interpretation in the light of earlier investigations, which appeared not to have been noted by the authors. The in vitro study has application and significance beyond the aims of the current study. In future in vivo measurements, it might be an advantage to adopt the Manley Robertson technique, which avoids some of the uncertainties encountered by the auditory nerve measurement technique used in the paper under review.

We appreciate the constructive suggestions from the reviewer and will adopt the recommended strategies in future studies.

Reviewer #3:The transduction of mechanical sound signals into electrical activities has so far been attributed to the inner hair cells of the inner ear. In this manuscript the authors show that the spiral ganglion neurons that innervate the inner hair cells are mechanosensitive as well. In particular, the authors find that spiral ganglion cells maintained in cell culture can fire action potentials in response to mechanical stimulation. Moreover, the authors provide evidence that the mechanotransduction in the spiral ganglion cells may contribute to sound detection in vivo. These results are overall well described and are supported by a variety of information and data.Some questions and concerns that I have are:1. The authors show that the channel blocker GsMTx4 reduces the eveoked current by roughly 50%. But this means that there are still further mechanotransduction channels that are not blocked by GsMTx4. Have the authors tried out other channel blockers to reduce the evoked current further?

We thank the reviewer for the suggestion and concerns. We showed that 1mM blocked ~50% of the mech current. On the other hand, we showed in the dose-response curve and summary data that higher concentrations of GsMTx4 blocked the mech current completely. To our knowledge, there is no mech channel-specific blocker besides GsMTx4. (Figure 1, page 48 line 1222).

2. The authors state that panels f and g in Figure 3 show the effects of peak splitting. However, I can't infer this information from the panels, since the spikes are so close together. I recommend the authors show the spikes in a much shorter time window, together with the stimulus waveforms, so one can compare the spike timings to the phases of the stimulations. I am also not sure how to read panels h and m, please describe the presented information in more detail.

We appreciate the reviewer's concern. Peak-splitting is revealed by cycle (or period) histograms next to (for Figure 3a-c) or below (for Figure 3d-k) the spike traces. We refer to these panels more explicitly now in the main text.

In these panels, the spike rate is plot as a function of stimulus phase. “Perfect” phase-locking (vector strength = 1) is obtained when all spikes are fired at the same phase-angle. Figure 3a,c,d,i.j approach that situation. Peak-splitting means that spikes are fired at multiple preferred phaseangles: this is observed in Figure 3b,e,f,g,k. Meanwhile, the other reviewers are concerned that we showed “too much data,” making the figures appear cluttered. We want to refer the reviewer to Figure S9, where we illustrated an expanded view to similar traces (Figure 3 page 50 line 1228; Figure S9, page 60 line 1274).

3. As I understand it, the spiking of the spiral ganglion neurons is independent of the direction of the flow respectively of the direction of the mechanical stimulation. As a consequence, when investigating the dependency of the firing rate on the phase between the current stimulation and the mechanical stimulus, a phase shift of 180 degrees should give the same result as no phase shift. In other words, the dependence on the phase shift should be periodic with a period of 180 degrees. Is that the case?

This is an interesting suggestion, but impossible to test. We do not know what the in vivo motion is that triggers or influences spiking, and not even whether it really is motion or rather a deformation (e.g. pressure differences at the habenula perforata). The in vitro experiments do not give that information either, since the mechanical stimulus is applied to the substrate on which the neurons are grown. While one can “push” the substrate with a glass probe, it is not possible to “pull”, and even if such pulling would be achieved it would not necessarily simulate the motion or deformation that an unmyelinated SGN dendrite experiences in vivo*.*

4. The biggest uncertainty to me is the relevance of the described mechanotransduction in vivo. The authors present results from in-vivo experiments in which they block the synaptic transmission between the inner hair cell and the spiral ganglion neuron. They show that the low-frequency part of the tuning curves remains pretty much unchanged, demonstrating that this portion of the tuning curve results from a second mechanontransduction process. But I have doubts that this second mechanotransduction process is the one that the authors have characterized before, due to the frequency response. After application of the synaptic blocker, the tuning curves are unchanged at frequencies up to 1kHz, although the authors have shown that the mechanotransduction in isolated spiral ganglion neurons was attenuated above 10 Hz and absent at 1 kHz. How can these findings be reconciled with an in vivo function up to 1 kHz?

We are unclear why the reviewer states that mechanotransduction in SGNs attenuates above 10 Hz. This was the case for vestibular ganglion neurons (Figure S5d) but not in SGNs, at least not uniformly. Figure S7 (e) shows examples of SGN mechanical responses at 1, 10, 100 Hz. For 2 of the 3 examples shown, response was best at 100 Hz. Also, there was a general increase in phase-locking (increase in vector strength VS) at 100 Hz relative to 10 Hz as shown in panel g of that figure (admittedly, there was a mislabeling between f and g in the legend of the figure of the original manuscript). Unfortunately, mechanical stimulation in vitro above 100 Hz was unreliable because of limitations in the actuator. Thus, the in vitro data unfortunately do not inform us re. mechanical effects at the higher frequencies tested in vivo. A second issue is related to the answer to the previous question. We do not know the exact nature of the mechanical stimulus in vivo. As acknowledged in the original manuscript, the in vitro stimulus is undoubtedly only a crude approximation of the in vivo situation.

5. It appears to me that a more direct test of the in vivo significance of the mechanotransduction in spiral ganglion neurons would be the application of the channel blocker GsMTx4 in vivo. The authors have shown that this blocker reduces the mechanotransduction in spiral ganglion neurons by approximately 50%. If the application of this blocker in vivo produced an elevation of the tuning curves, this would provide further evidence for the involvement of this pathway in in-vivo sound detection.

We have thought about the suggested experiments. The experimental conditions did not allow for the perfusion and diffusion of large-molecular weight peptides in the intact cochlea in vivo experiments. (Please see references cited Verschooten et al., 2015; Sadreev et al., 2019).

6. The last sentence regarding new treatments for hearing-impaired persons appears out of the blue. The authors should either eliminate this sentence, or explain further how such new treatments could work.

The last comment contrasts with Reviewer 1, who states: the last sentence of the discussion is the most important. We have therefore maintained and expanded the statement (page 23 lines 571-575).

[Editors’ note: what follows is the authors’ response to the second round of review.]

Essential revisions:1. 93-100 AN responses, which generate unimodal cycle histograms i.e., response as a function of stimulus phase (Johnson 1980, Rose et al. 1967). In contrast, for some intermediate intensities, AN fibers fire APs at two or more stimulus phases, a phenomenon referred to as "peak-splitting" (Johnson 1980, Kiang and Moxon 1972). Moreover, a typical AN frequency tuning curve consists of two components (Liberman and Kiang 1984) – a sharply tuned tip near the characteristic frequency (CF) and a low-frequency tail, which are differentially sensitive to cochlear trauma. Both of these observations suggest that more than one process may drive AN responses.Comment: These topics are covered by recent papers (e.g Hear Res. 2016; 341: 66-78; Hear Res. 2018 Feb;358:1-9. doi: 10.1016) that involve presynaptic mechanism. Perhaps these papers should be quoted here and commented on in the discussion.

We added these references in the section where we discuss underlying mechanisms (Discussion, section "Mutually interacting elements of neurotransmission and mechanical activation at the first auditory synapse"); (see page 4 line 92 and page 21 line 516-517).

2. Lines 108-111The geometry of the course of the unmyelinated terminal segment of SGN dendrites towards the Organ of Corti (OC) suggests that this segment undergoes some degree of mechanical deformation in response to sound.Comment: This is an important point. Please quote supporting reference.

We added a sentence to expand on this point and added a reference to Lim (1986): (see page 5 line 109-110).

3. Lines 137-138. "subject to OC movement (OCM) (Chen et al. 2011, Jawadi et al. 2016, Karavitaki and Mountain 2007),Comment: I have read these papers and could not see where any of the authors states explicitly that the nerve terminals are subject to OC movement. Please correct me, if I am wrong, but it appears that you have deduced this without proof. And should make clear that this is your deduction and not a finding reported in the papers. I can see mention of previous measurements of radial motion of MOC fibres during OHC contractions but the resolution of the method, stated by the authors was too weak to resolve the very small movements they observed near the edge of the reticular lamina.

Cited reference indeed does not state that unmyelinated SGN terminals are subject to OC movement. We now state that it was inferred from a supplementary video.

The link below shows the video that Karavitaki and Mountain 2007, provided as supplementary material. The video legend states: "Movie2: this movie shows the electrically evoked micromechanical motion of the organ of Corti when we focused at the basal end of the OHCs. The stimulus frequency is 90 Hz." https://ars.els-cdn.com/content/image/1-s2.0-S0006349507711360-mmc2.avi

4. Lines 189-191 The mechanical responses of VNs were comparable to those reported for dorsal root ganglion (DRG) neurons (Finno et al. 2019, Viatchenko-Karpinski and Gu 2016).Comment: I am not sure. Your figure 1A shows events that appear not to be all-or-nothing spikes

We are unsure which point the reviewer is making here. Figure 1A shows responses from SGNs, not VNs. The time scale of Figure 1A is very compressed, showing more than 10 min. of data. The events that look smaller than APs are depolarizations that decay in amplitude.

Figure S5C represents responses from VNs. Traces corresponding to 1Hz and 1000Hz also show APs that decay in amplitude during the current injection.

5. Lines 227-229 Since the exact relationship in amplitude and phase between synaptic and mechanical events in the cochlea is unknown.Comment: The phase relationships are known between SPL, middle ear, BM, RL, Hair cell and neural responses, for example, and the IHC synaptic delay has been measured in vivo to be around 1ms. If the nerve fibres were excited directly by sound stimulation, they might be expected to respond in phase with the IHC responses and not as observed for moderate to high level (Palmer and Russell, 1986 and see Rutherford et al., 2021), after a ~1ms delay.

We agree that there are estimates of these phase and time relationships, although we do not think they are empirically known with the accuracy that the reviewer seems to imply. The ~1 ms is a convenient estimate (e.g., Ruggero and Rich 1987: includes travel time synapse – recording site), but we are not aware of in vivo measurements of IHC synaptic delay, which at present is experimentally impossible as it would require paired IHC and nerve recordings in a life animal. At the level of neural responses, which we have studied extensively in cat, the response phase at low frequencies shows very complex and idiosyncratic behavior which we have not been able to understand (and which therefore remains unpublished except in abstract form: van der Heijden and Joris, 2006) but which suggests a much more complex picture than the published literature. However, we agree with the reviewer that we misstated and overstated our main point. The key point is the relationship between synaptic and direct mechanical effects on the dendrites. We see no reason to expect that direct mechanical effects on dendrites would be in-phase with the IHC response. The difficulty is that the exact nature of the mechanical stimulus in vivo is unknown: is it the actual movement of the dendrite (and if so, which 3D component is most important?) or is it a deformation e.g., by a pressure gradient across the foramina? Moreover, even in the in vitro experimental conditions, where we have control over electrical and mechanical stimuli, the interaction between the two differs between cells (see Figures 3 and S11). We rephrased our statement to make this point more clear.

6. Line 295-377: Auditory nerve responses in the absence of synaptic transmissionComment: I still find this section unconvincing evidence in support of AN responding to mechanical stimulation in the absence of synaptic transmission. There remains the possibility, raised previously, that the applied agents have not fully blocked afferent transmission because they have not reached the location of the synapse. This situation is exacerbated by another recent report (J Neurophysiol. 2019 Mar 1;121(3):1018-1033). In this paper the authors show that: "as sound level is increased, the cochlear origins of CAPs from tone bursts of all frequencies become very wide and their centers shift toward the most sensitive cochlear region". Although you measure single units, this finding makes the interpretation very difficult. It is more reason to make measurements from the spiral ganglia directly.

We like to reverse the argument: it is precisely because the CAP (which we have also extensively studied and are intimately familiar with, e.g. Verschooten et al. 2012, 2018) is a population measure that it is very hard to interpret. This is not a new insight but has been abundantly clear since the first studies into the CAP's origin (e.g. Antoli-Candela and Kiang, 1978; Kiang et al., 1976). Single units do not suffer from this specific difficulty.

It is not clear to us why nerve recordings via the spiral ganglion technique would necessarily provide stronger evidence. First, we like to point out that this technique is inherently invasive to the cochlea, which is not the case for recordings from the nerve trunk, and that such recordings have a much lower yield. Second, in a given animal, the tonotopic sequence of fibers encountered when recording from the nerve trunk is reproducible across repeated tracks, particularly when the recording electrode is left in situ. Admittedly, this is only rarely documented in the published literature (see e.g. Liberman and Kiang, 1978). It is difficult to argue that all the low-frequency-dominated tuning curves observed in the initial part of the recordings of Figure 4, after administration of blocker, would be derived from apical fibers when the preceding track pre-drug clearly shows fibers tuned to high CFs. The key point is the *differential* effect on tip and tail. If the tuning curves obtained after administration of blocker would only reflect a partial effect of blocker, as suggested by the reviewer, why would that partial synaptic block affect the tip much more dramatically than the tail? We added a statement to clarify this expectation at the end of the first paragraph of this section as well as in the description of Figure 4a-f.

7. Lines 305-307 Cochlear displacements of this magnitude have been measured in vivo, particularly towards the cochlear apex and at high sound pressure levels (SPLs), where frequency tuning is poor (Cooper and Rhode 1995, Lee et al. 2015, Lee et al. 2016a).Comment: One should remember that a loud tone at the apex causes a much larger mechanical displacements than at the base of the cochlea and that loud tones in cochleae that are sensitive to ultrasound move far less for a given SPL than in the same spatial region of a low frequency cochlea eg gerbil. You are correct in saying that it is important to understand the nature of any mechanical stimulus that would excite ANs in vivo.

If we understand these comments correctly, we agree with the reviewer and believe that this is the point we were trying to make (though in softer form) in the manuscript, i.e., particularly for low-frequency sounds, to which peak-splitting has been observed, large displacements have been measured. When writing our manuscript, we actually inquired regarding a more general formulation (larger maximal displacements in apex vs. base) with colleagues performing cochlear mechanical measurements. The difficulty of obtaining high-quality apical mechanical measurements makes it somewhat tenuous to make a strong sweeping statement.

8. Lines: 483-489 Peak-splitting suggests an interaction of two pathways with different growth functions, which sum at the level of the AN (Kiang 1990, Kiang et al. 1986). We observed phenomenologically similar events in vitro when combining sinusoidal current injection and displacement at varying relative phases. Some combinations generated peak-splitting, accompanied by decreases in firing rate (Figure 3, S10). These findings suggest that the mechanical sensitivity of SGNs should be considered a possible factor in the suppressive and peak-splitting phenomena observed in vivo.Comment: In addition to Kiangs hypothesis, peak splitting has recently been attributed to presynaptic mechanisms (Nam and Guinan, Hear Res. 2016 November ; 341: 66-78.). There is support for this idea from IHC intracellular measurements, but not OHCs (Russell and Kössl, 1992, Figure 4), although these findings were obtained from measurements whose objective was not to examine the basis of Nelson's notch.

The study of Russell and Kössl (1992) reports the effect of a bias tone (100 Hz) on responses of high-CF hair cells. Indeed, for such combined stimuli, IHCs show responses with a strong 2^nd^ harmonic, depending on the level of the high-frequency tone. The relationship to peak-splitting is unclear as discussed in our manuscript (evoked by single tones). Indeed, Russell and Kössl explicitly discuss their findings as an effect of the *interaction* between the low-frequency bias tone and the high-frequency tones, and do not show or discuss responses to single low-frequency tones. There is a large literature on the effect of bias tones (always of *very* low frequency) on responses to other stimuli in the auditory nerve. Similar to the Nam and Guinan papers referenced by the editor/reviewer (which don't address the Russell and Kössl data), bias tones are introduced in order to evoke slow positional biases of the basilar membrane towards scala media or scala tympani.

However, we take the general point of the editor/reviewer: note that we already acknowledged the possibility raised (presynaptic origin of peak-splitting) in the previous version of the manuscript (Discussion, section "Mutually interacting elements of neurotransmission and mechanical activation at the first auditory synapse"). We added several more references, including the study by Russell and Kössl, to our statement. We think a fair summary of the data in the literature is that the origin of peak-splitting remains a contested topic, with many different types of data hinting at several possibilities. Since our in vivo experiments do not provide data addressing peak-splitting and our statements regarding this phenomenon are only based on the in vitro results, we are reluctant to make strong statements, except to point out that direct mechanical effects on AN dendrites would provide a straightforward and hitherto unrecognized "second path".

9. Lines 509-514: Similar phenomena, particularly regarding phase-locking, have been reported in other species but were not always restricted to high sound levels. The source of these components is controversial but has been sought at the level of cochlear mechanics or hair cells, not the AN (Cai and Geisler 1996, Cody and Mountain 1989, Dallos 1985, Heil and Peterson 2019, Kiang 1990, Liberman and Kiang 1984, Ruggero and Rich 1983, Ruggero et 514 al 1986).Comment: See above and perhaps include the Nam and Guinan, 2016 paper.

References added.

10. Lines 516-523: We hypothesized that intrinsic mechanical sensitivity contributes to the high-threshold, "tail" region of tuning curves, which is less vulnerable to a range of cochlear manipulations than the tip (Kiang et al. 1986, Liberman and Kiang 1984). These tails may be associated with clinically relevant phenomena, such as recruitment of middle ear reflexes and abnormal growth of loudness after acoustic trauma (Liberman and Kiang 1984). Delivery of synaptic blockers diminished or abolished spontaneous activity and the tip region of the tuning curve. The observation that AN responses persisted at high sound levels is in keeping with the prediction that a second mode, direct mechanical activation of AN fibers, operates and dominates high threshold segments of the tuning curve (Figure 4, S12).Comment: See Nam and Guinan Hear Res. 2018 February; 358: 1-9. Who suggest that changes due to low frequency tail are due to the shearing reticular membrane motion

The experimental data in the pair of papers by Nam and Guinan also consist of recordings from the auditory nerve, and their interpretation of the two components of frequency tuning is necessarily speculative, as is our interpretation as well of that of others (e.g. Liberman and Kiang, 1984) based on such recordings. We believe the value of our contribution is that, even though our data are necessarily indirect, they introduce a completely new and unexpected angle to such questions.

11. Lines 528-538:Comment: Please see comment above to line 295-377

The discussion on lines 528-538 just makes the point that the two modes of synaptic release observed between IHC and AN fibers (Grant et al. 2010) do not provide an explanation of the dual nature of the AN tuning curves. This point is not made in the section of (preceding) lines 295-377.

12. Figure 3: the phase locking shown in panels h and m does not appear terribly strong. In particular, the distributions do not differ dramatically from a uniform one, although the differences are probably significant. To be on the safe side, I would therefore like to see a statistical test that the distributions are indeed significantly non-uniform.13. Figure 3: The authors state that "the close alignment of the two lines shows that a high firing rate is accompanied by strong phase-locking, and vice versa that peak-splitting is associated with low firing rates." I don't understand how I am suppose to see that. First, as pointed out above, phase locking looks rather weak to me. Second, where are the two cases (high and low firing rates) represented?14. Figure 3: Connected to the above, since panels h and m summarize data with and without peak splitting, shouldn't one see an effect of peak splitting in these histograms? In other words, how can peak splitting be inferred from the histograms?

These 3 comments (12,13,14) all seem to reflect a misreading of the figure. We have considerably expanded our description of the figure in the main text and figure legend.

The panels with polar plots are not intended to illustrate the strength of phase-locking per se, but rather the relation between firing rate and phase-locking strength. Two polar plots are superimposed: for firing rate (blue) and VS value (red). The phase angle in the polar plot gives the *relative* stimulus phase between mechanical and electrical stimuli (angle is the phase lead of current over mechanical stimulus). For example, in panel m there is very high firing and exquisite phase-locking (VS ~ 1) at a phase difference of 90 degrees (see also Figure 3j). When the stimuli are out-of-phase (180 degrees), the firing rate diminishes relative to that at 90 degrees, and phase-locking also drops to a value lower than 1. The period histogram of the response (Figure 3k, lower panel) shows that this is due to peak-splitting. With a further phase current phase led to 270 degrees, the cell stopped firing.

In general, in panels h and m, as well in the further examples provided in supplementary figure S11 (left), there is a covariation of firing rate and strength of phase-locking: when firing rate is high, VS tends to be high, and vice versa. Because the phase between electrical and mechanical stimulation is a circular variable, a simple way to illustrate the covariation is to calculate a vector average of the two polar plots. The phase angle of this resultant rather than its magnitude is of interest and shown with the heavy lines in the polar plots: the similarity in angle between rate and VS illustrates that a decrease in rate accompanies peak-splitting. For those cases where responses to a full circle of phase-angles between the two stimuli were not available, we performed the analysis of Supplementary Figure 11 (right panel).

To complete the figure, particularly in response to comment 12, we added a table (1C) for the 5 cases where a full circle of relative phases was tested (2 cases from Figure 3, 3 cases from Figure S11). The table lists the maximal value of firing (used to normalize the rate responses), and (for both rate and phase-locking) the magnitude and phase of the resultant vector, as well as the p value of the Rayleigh test. All polar plots were statistically different from a uniform distribution (p < 0.005).

15. Lines 82-87: Phase locking to sound stimuli is a feature of the AN essential for sound detection, localization, and arguably for pitch perception and speech intelligibility (Peterson and Heil 2020, Yin et al. 2019). How these response features remain sustained, despite the limits of presynaptic mechanisms of transmitter release to ATP-generation, synaptic fatigue, and vesicle replenishment (MacLeod and 86 Horiuchi 2011, Stevens and Wesseling 1999, Yamamoto and Kurokawa 1970), is not fully understood.Comment: Perhaps include the excellent review by Rutherford et al., which covers this topic (J Physiol 599.10 (2021) pp 2527-2557).

Reference added.

Reference:

Antoli-Candela, E.J., Kiang, N.Y.S., 1978. Unit activity underlying the N1 potential, in: Evoked Electrical Activity in the Auditory Nervous System. Academic Press, New York, pp. 165–189.

Kiang, N.Y.S., Moxon, E.C., Kahn, A.R., 1976. The Relationship of Gross Potentials Recorded from the Cochlea to Single Unit Activity in the Auditory Nerve, in: Electrocochleography. University Park Press, Baltimore, pp. 95–115.

Liberman, M.C., Kiang, N.Y., 1978. Acoustic trauma in cats. Cochlear pathology and auditory-nerve activity. Acta Otolaryngol Suppl 358, 1–63.

Lim, D.J., 1986. Functional structure of the organ of Corti: a review. Hearing Research 22, 117–146. https://doi.org/10.1016/0378-5955(86)90089-4

Nam, H., Guinan, J.J., 2018. Non-tip auditory-nerve responses that are suppressed by low-frequency bias tones originate from reticular lamina motion. Hear Res 358, 1–9. https://doi.org/10.1016/j.heares.2017.12.008

Nam, H., Guinan, J.J., 2016. Low-frequency bias tone suppression of auditory-nerve responses to low-level clicks and tones. Hearing Research 341, 66–78. https://doi.org/10.1016/j.heares.2016.08.007

Ruggero, M.A., Rich, N.C., 1987. Timing of Spikes initiation in Cochlear Afferents: Dependence on Site of Innervation. J Neurophysiol 58, 379–403.

Rutherford, M.A., von Gersdorff, H., Goutman, J.D., 2021. Encoding sound in the cochlea: from receptor potential to afferent discharge. J Physiol 599, 2527–2557. https://doi.org/10.1113/JP279189

van der Heijden, M., Joris, P.X., 2006. Effects of stimulus intensity on phase and amplitude characteristics of auditory nerve fibers. Assoc Res Otolaryngol Abs 29, 28–29.

Verschooten, E., Desloovere, C., Joris, P.X., 2018. High-resolution frequency tuning but not temporal coding in the human cochlea. PLOS Biology 16, e2005164. https://doi.org/10.1371/journal.pbio.2005164

Verschooten, E., Robles, L., Kovačić, D., Joris, P.X., 2012. Auditory nerve frequency tuning measured with forward-masked compound action potentials. J. Assoc. Res. Otolaryngol. 13, 799–817. https://doi.org/10.1007/s10162-012-0346-z